# Mend the Gap: Online User-Led Adjuvant Treatment for Psychosis: A Systematic Review on Recent Findings

**DOI:** 10.3390/ijerph22071024

**Published:** 2025-06-27

**Authors:** Pedro Andrade, Nuno Sanfins, Jacinto Azevedo

**Affiliations:** 1Faculty of Medicine, University of Porto, 4200-319 Porto, Portugal; nunomanuelsanfins@gmail.com; 2Neuroscience Department of Hospital Central Vila do Conde, Grupo Trofa Saúde, 4480-565 Vila do Conde, Portugal; jacintoazevedo@gmail.com; 3RISE-Health, Department of Clinical Neurosciences and Mental Health, Faculty of Medicine, University of Porto, Alameda Prof. Hernâni Monteiro, 4200-319 Porto, Portugal

**Keywords:** online, user led, intervention, schizophrenia, psychosis, treatment

## Abstract

**Background/Objectives:** Schizophrenia Spectrum Disorders (SSDs) carry a debilitating burden of disease which, even after pharmacological and psychological treatment are optimized, remains difficult to fully target. New online-delivered and user-led interventions may provide an appropriate, cost-effective answer to this problem. This study aims to retrieve the currently gathered findings on the efficacy of these interventions across several outcomes, such as symptom severity, social cognition, functioning and others. **Methods:** A systematic review of the current available literature was conducted. Of 29 potentially relevant articles, 26 were included and assigned at least one of four intervention types: Web-Based Therapy (WBT), Web-Based Psycho-Education (WBP), Online Peer Support (OPS) and Prompt-Based Intervention (PBI). **Results:** The findings were grouped based on outcome. Of 24 studies evaluating the effects of symptom severity, 14 have achieved statistically significant results, and 10 have not. WBT (such as online-delivered Cognitive Behavioral Therapy, Acceptance and Commitment Therapy, social cognition training and Mindfulness Training) seemed to be the most effective at targeting symptoms. Of 14 studies evaluating functioning, seven achieved significant results, four involving a form of social or neurocognitive training, suggesting a potential pathway towards functional improvements through interventions targeting cognition and motivation. Regarding social cognition, all seven studies measuring the effects of an intervention on this outcome produced significant results, indicating that this outcome lends itself well to remote, online administration. This may be linked with the nature of social cognition exercises, as they are commonly administered through a digital medium (such as pictures, videos and auditory exercises), a delivery method that suits the online-user led model very well. **Conclusions:** Online user-led interventions show promise as a new way to tackle functional deficits in SSD patients and achieve these improvements through targeting social cognition, a hard-to-reach component of the burden of SSDs which seems to be successfully targetable in a remote, user-led fashion. Symptomatic improvements can also be achievable, through the combination of these interventions with treatment as usual.

## 1. Introduction

Schizophrenia Spectrum Disorders (SSDs)—which include brief psychotic disorder, schizophreniform disorder, schizoaffective disorder, schizophrenia or psychotic disorder not otherwise specified—are some of the most detrimental conditions to a patient’s quality of life, as they are a life-long endeavor punctuated with the constant threat of acute episodes that have the potential of endangering both the patient’s and other people’s safety, as well as lead to prolonged hospitalization, incurring significant costs. Both chronically and acutely, these SSDs are treated systematically via antipsychotic medication, as well as some variations of psychotherapy [1].

Active involvement in treatment plays a major role in improving patients’ outcomes, especially in the context of schizophrenia. Non-adherence to treatment as prescribed has long been significantly associated with the rate of relapse and hospitalization and proven to be an indicator of poor clinical outcomes [2].

Although schizophrenia faces a similar challenge to other mental illnesses regarding a treatment gap—the absolute difference between the true prevalence of a disorder and the treated proportion of individuals affected by the disorder—this gap has shown to be narrower, at about 32.2%, when compared to other serious mental illnesses (depression at 56.3% and bipolar disorder at 56%) [1].

Thus, the gap that is the subject of focus in this study will be a therapeutic one—defined as the overall burden of disease left unchanged even after pharmacological and psychotherapy intervention have been optimized. Contrary to the treatment gap, which draws from the total number of SSD patients (currently in treatment or not), the therapeutic gap is seen only among SSD patients in active treatment. Additionally, the therapeutic gap addresses specific clinical outcomes within the broader burden of disease of SSDs, whereas the treatment gap only refers to the number of patients in treatment (regardless of how effective it may be in improving these different outcomes).

Antipsychotic medication, as well as various models of psychotherapy (such as CBT and ACT) have shown to produce great long-term improvements, such as a reduction in the frequency and severity of relapse and suicidality, up to 60%, as well as cognitive and functioning improvements. Additionally, pharmacotherapy also plays the most significant part in treating acute psychosis, as well as early stages of illness, contributing to a reduction in symptom severity, in up to 85% of patients [3,4].

However, some aspects of the burdens of SSDs are left unanswered by these treatments, such as negative symptoms (avolition, alogia and expressive deficits) afflicting nearly 60% of patients [5].

The current pharmacological offer for SSDs, mainly focused on a dopamine receptor blockade, has yet to show significant efficacy in approaching these symptoms [5], indicating a need for new modalities of treatment more suited for this aspect of SSD, which has been described in some studies as “the holy grail of treatment research in schizophrenia” [6].

Psychosocial interventions have shown to be effective in augmenting pharmacological and psychological treatment and improving patient outcomes, as well as reaching clinically significant improvements in those “difficult to treat” components of the illness. These were mainly achieved through indirect pathways, connected to social cognition and motivation [6].

Despite this, the rate of penetration for these interventions has hindered their potential in addressing the therapeutic gap for schizophrenia, with rates measured around less than 10%. This could be attributed to the highly specialized nature of these interventions, as they employ treatment modalities (such as CBT, ACT and others) that require specialized training of mental healthcare workers, and incur significant time and monetary expenditure to be successfully administered. Geographic barriers and difficulty in transportation may also prevent patients from accessing this kind of treatment, hindering attendance and procurement [7]. Additionally, face-to-face administration of these interventions in a routine clinical setting may also contribute to the current over-burdening of mental health workers, pointing to a need for a new and modern solution to this problem [8].

In fact, schizophrenia patients require a long-term and active management approach of their symptoms and on-going treatment, if sustained improvements in clinical outcomes are to be achieved. This calls for an intervention which is (a) cost effective, (b) non-stigmatizing, (c) unrestricted by time and (d) offered beyond discharge from mental health services [7,9].

All of the above-mentioned requirements are met by recently developed, online, user-led interventions, which integrate these services within a widely propagated and consistently frequented environment—the Internet.

These interventions may be the solution to bridge the therapeutic gap, in a way that is feasible and accessible to healthcare providers and patients, as they benefit from greater flexibility in the way they are administered, and require less extensive training of mental health providers to be useful [8,10,11].

In support of the potential of these interventions, recent studies have shown that general Internet use in patients with schizophrenia is comparable to that of the general population, and even more so as a source of health information and coping strategy [12]. Access to the Internet has also shown to be comparable [13,14].

In light of this, over the last 20 years, many interventions have been formulated, and accessed in terms of their feasibility and acceptability, as well as preliminarily tested in pilot trials. Despite this, the overall knowledge of their efficacy and potential for prescription as adjuvant treatment for SSD has been scarce, and, to our knowledge, there have been only two systematic reviews targeting the full scope of available online, user-led interventions for SSDs [7,15] and two specifically targeting smartphone apps [16,17]. Only one of these studies, in 2016, has made an attempt at a meta-analytic evaluation of these results [14].

In the last decade since these reviews, there has been a significant growth in the research, with multiple new interventions being developed, piloted [18,19,20] and subsequently evaluated in increasingly robust clinical trials [8,21].

This calls for a new assessment of the available literature. Since the feasibility and usability of Internet-delivered psychosis intervention has been amply demonstrated across most studies [22,23], the main focus of this paper will be on the efficacy of these interventions.

Where, in the previous decades, steps have been taken to highlight and hypothesize solutions for the unattended aspects of schizophrenia treatment—Minding the Gap—this decade will be focused on developing these solutions and optimizing our approach to this problem using the Internet so that, in the future, online user-led interventions for the treatment of psychosis may be available for prescription alongside pharmacological and psychological modalities—Mending the Gap.

In this study, we aim to gather the current available research on online, user-led interventions for Schizophrenia Spectrum Disorders (SSDs) and perform a narrative review of our findings.

## 2. Materials and Methods

This systematic review was conducted in accordance with the Preferred Reporting Items for Systematic Reviews and Meta-analysis (PRISMA) [24]. This study has been submitted for registration for the PROSPERO database for systematic reviews (registration code CRD420250632405).

Three databases (PubMed, Scopus and ISI Web of Science) were searched for potentially relevant abstracts with the aim to attain sufficient representation of all publications across all medical fields on this subject. In our search specifications we included published (or in-press) English-only manuscripts. No time frame was specified.

The search term used was the same for all 3 databases and focused on ensuring a broad coverage of studies on online interventions in our review. The following search terms were used: (“Schizophrenia” OR “Psychosis”) AND “Outcomes” AND (“Internet” OR “Online” OR “Social Media” OR “Web-based” OR “Website” OR “User-led”). For PubMed specifically, we did not use MeSH terms to formulate our search query. We limited the scope of our review to publications in English with an accessible abstract.

We included studies whose participants were people with Schizophrenia Spectrum Disorders (SSDs)—brief psychotic disorder, schizophreniform disorder, schizoaffective disorder, schizophrenia or psychotic disorder not otherwise specified—or other serious mental illnesses (SMIs) diagnosed by a qualified mental health professional using either DSM or ICD criteria. If the study included participants with other SMIs, the study was included if the sample was composed of at least 70% SSD patients. Participants of all ages were considered for inclusion.

The topic of the included studies had to concern online, user-led interventions for SSD. We adopted a previous definition of online interventions as web-based interventions enabling peer-to-peer contact, patient-to-expert communication or interactive psycho-education/therapy [7]. User-led interventions were likewise described as interventions in which participants led or directed the timing, content or interaction with the web or mobile-based intervention [7].

Studies exclusively investigating traditional face-to-face therapy delivered via teleconference using mobile phones were excluded, except when teleconference was not the only component of treatment, and merely a facet of a multi-layered, and otherwise independent treatment plan, and only when delivered for the purposes of group therapy, as our goal is to seek interventions that reduce the amount of mental health personnel required to effectively deliver these interventions.

Studies were excluded if the intervention was used exclusively to collect data (e.g., online surveys, electronic medical records and digital phenotyping) with no additional online social networking, or usage of such data to customize or otherwise deliver the intervention. Included papers could, however, dedicate a section to data collection alongside their main findings on a relevant intervention.

We have also excluded papers focused on the design or the development process of the intervention. Theoretical or methodological papers, books or book chapters, letters, dissertations, editorials or study protocols were excluded.

In terms of comparators, any control group was accepted as long as it was composed of other SSD patients (at least 70%), either receiving treatment as usual (TAU) or other interventions (both outcome-specific like cognitive remediation therapy [25], and outcome-unspecific like progressive muscle relaxation exercises [26] and conventional videogames [27]). Studies in which the comparator group was a healthy population were excluded. As previously stated, our goal is to tackle the therapeutic gap within the SSD population, as opposed to the treatment gap in those undiagnosed among the general population.

As we intended to cast a broad net onto the current landscape on this topic, no exclusion criteria were set for outcomes. Any outcome directly or indirectly related to SSD was to be extracted. Relevant outcomes were only selected and narratively described after data extraction.

The study type was limited to randomized controlled trials as well as single-arm pre–post trials, provided they measured clinical outcomes for the intervention. Studies purely devoted to accessing feasibility and acceptability were excluded.

Two authors (P.A., N.S.) accessed the eligibility of all 4921 hits and excluded 4816 based on title, leaving 105 abstracts. Then, the same two authors screened these abstracts independently for eligibility, excluding ineligible records (*n* = 76). The remaining 29 articles were collected for detailed evaluation based on inclusion and exclusion criteria, with 13 being excluded, and 16 being included. The reasons for exclusion are detailed in Figure 1. Additionally, 10 papers were collected and included after a hand search of references from the selected papers was conducted. The final count for included studies was 26. In cases of discrepancy concerning the decisions made between the reviewers, the papers were discussed until a consensus was reached with the support of J.A. Figure 1 outlines the search process of the literature, according to PRISMA [24].

Firstly, data extraction was based, when possible, on the CONSORT-EHEALTH checklist, and data was collected using a matrix developed for gathering the following information: characteristics of the study (title, main author, country of origin, aim and design), intervention and control group sample sizes, demographic characteristics, attrition rate and duration of intervention follow-up period (whichever was the longest time elapsed from baseline to the last measurement phase).

Secondly, we attributed to each study at least one of four non-mutually exclusive intervention types: Web-Based Therapy, Web-Based Psycho-Education, Online Peer Support and Prompt-Based Intervention.

Thirdly, we registered the primary and secondary outcomes measured in each study. This was to identify which outcomes are most commonly targeted when developing these interventions. Additionally, we registered all instruments used to gather these measures.

The Revised Cochrane risk of bias tool for randomized trials [28] was employed separately by 2 reviewers (P.A., N.S.) to assess the quality of the included studies. This tool assesses risk of bias in 5 domains, each addressing a possible source of bias: the randomization process, deviations from intended interventions, missing outcome data and selection of the reported result as well as overall risk of bias. A visual representation of this appraisal can be consulted in Figure 2.

For uncontrolled studies, we assessed 3 criteria: the integration and nature of blinding into the study design, the existence or not of outcome data and retention/attrition of the intervention assessed against a priori criteria. Under these criteria, all 5 uncontrolled studies were deemed to have a low to medium risk of bias.

In case of disagreement, different views were discussed amongst all authors (P.A., N.S., J.A.).

The main concern over the methodological quality of included studies lies in Domain 2: bias due to deviations from intended interventions, particularly in the concealment of the intervention from the participants in the experimental group. The user-led nature of these interventions, which are often delivered through websites, applications and content that the participants access on their own terms, leads to participants having an innate knowledge that they are assigned to the experimental group. This is especially relevant considering that a number of outcomes (such as motivation, self-efficacy and others) were measured using self-assessment scales, so placebo can ultimately overstate some of the results. Despite this, most studies conducted evaluation of outcomes through a blinded, unbiased assessor, which works to mitigate this methodological concern. A minority of unblinded studies [23,29] contributed to the majority of concerns regarding risk of bias, as can be seen in Figure 2.

For narrative analysis, data on each included study was manually categorized and entered into the data extraction matrix. A diagram was made, grouping all included interventions according to their intervention types. A more detailed description of each of the four treatment modalities was made.

The descriptive analysis was centered around outcome, and so studies were presented within each section in regards to their results for that specific outcome. This was made to provide a more in-depth view of the current landscape of research, and identify which areas have shown promise, and which have not.

## 3. Results

Of the 29 studies retrieved for detailed evaluation, 16 were fit for inclusion in the present study. Additionally, 10 studies were retrieved from the reference list of the evaluated studies, when they fit the inclusion criteria.

### 3.1. General Characteristics of the Included Studies

#### 3.1.1. Population and Attrition

A total of 26 studies, in which participated 2372 patients diagnosed with either SSD or first-episode psychosis (FEP), were included. Out of the 26 studies, only one tested an intervention on a population of FEP patients. The mean age varied between 20.91 and 51.2 years. Attrition rates varied greatly from 0% to 70.8%. There was an overlap of participants between two of the included studies [6,21], one being an interim report conducted immediately after the intervention was administered [6], and another measuring the same outcomes on the same population after a 6-month follow-up period [21]. There was no participant or data overlap between any of the remaining studies.

#### 3.1.2. Design

Of the included studies, five employed a single-armed, pre–post design [20,30,31,32,33]. The remaining 21 studies employed a controlled trial design, of which 20 were randomized and one unrandomized [29]. One of the included randomized controlled trials (RCTs) was three-armed [34], as opposed to all other studies, which were parallel-armed. Among all RCTs, 11 were single-blinded, five were double-blinded [6,21,27,35,36] and four were unblinded [23,25,29,37]. Follow-up time ranged from 6 weeks to 18 months. Of the 21 controlled studies, the comparator group was assigned to an active control condition in seven cases [6,21,25,26,27,35,36], and to a wait-list condition in four cases [29,38,39,40]. The remaining 10 studies used treatment as usual (TAU) as a comparator to the intervention [8,23,34,37,41,42,43,44,45,46]. In one of these studies, TAU was supplemented with an educational video [43].

**Figure 2 ijerph-22-01024-f002:**
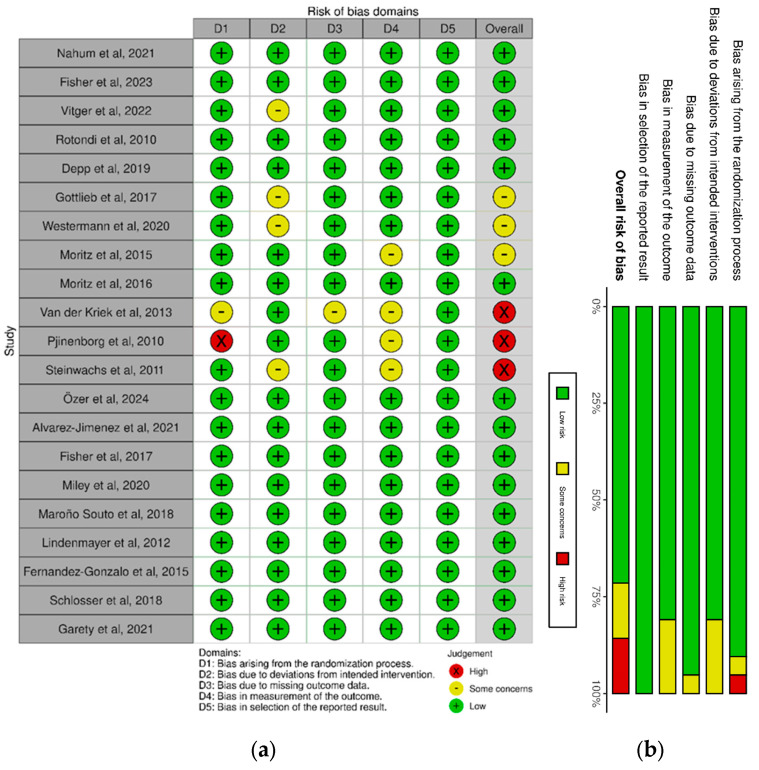
(**a**) Risk of bias domains for each controlled study; (**b**) overall risk of bias. References: Nahum et al., 2021 [27]; Fisher et al., 2023 [35], Vitger et al., 2022 [44], Rotondi et al., 2010 [37], Depp et al., 2019 [34], Gottlieb et al., 2017 [8], Westermann et al., 2020 [38], Moritz et al., 2015 [26], Moritz et al., 2016 [39], van der Krieke et al., 2013 [23], Pjinenborg et al., 2010 [29], Steinwachs et al., 2011 [43], Özer et al., 2024 [46], Álvarez-Jimemez et al., 2021 [42], Fisher et al., 2017 [6], Miley et al., 2020 [21], Maroño Souto et al., 2018 [41], Lindenmayer et al., 2012 [25], Fernandez-Gonzalo et al., 2015 [36], Schlosser et al., 2018 [40], Garety et al., 2021 [45].

#### 3.1.3. Origin

Twelve of the studies took place in the USA [6,8,20,21,25,27,33,34,35,37,40,43], three in Germany [24,38,39], two in UK [30,45], two in the Netherlands [23,29], two in Spain [36,41], two in Canada [31,32], one in Australia [42], one in Turkey [46] and one in Denmark [44].

### 3.2. Types of Intervention

#### 3.2.1. The Four Types of Online User-Led Interventions for SSD

In this review, we have attributed to each study at least one of four intervention types: Web-Based Therapy, Web-Based Psycho-Education, Online Peer Support and Prompt-Based Intervention. These are modified from the intervention types presented in a previous systematic review on the same topic (Alvarez-Jimenez et al., 2014) [7]. A single intervention can, however, qualify for all four simultaneously.

*Web-Based Therapy (WBT)* is defined in this study as any intervention that provides or otherwise facilitates access to treatment directed at specific symptoms/burden of SSD. These include Cognitive Behavioral Therapy [30,34], or CBT, Action/Commitment Therapy [46], or ACT (delivered through websites, mobile applications or teleconference only when delivered through group therapy), Self-Help, Mindfulness Training [28], Cognitive Remediation [25] and Auditory and social cognition training [6,21]. These can be delivered in various multimedia formats, such as visual and auditory exercises, as well as in the form of text, videos worksheets, images and illustrations delivered remotely via mobile device or a computer connected to the Internet. Some of the WBT interventions offer personalized treatment [34], based around self-reported symptoms and treatment goals, after an entry survey is conducted, or after an initial consultation.

*Web-Based Psycho-Education (WBP)* is defined in this study as any intervention that seeks to make information about symptoms, shared decision-making, self-management, treatment adherence, mindfulness and other useful knowledge around SSD available to the patient in a remote, user-led form [34,43,44]. Like WBT interventions, some WBP interventions provide means to personalize the content delivered [34].

*Online Peer Support (OPS)* is defined in this study as interventions that provide any online space, forum or social platform where SSD patients and relatives/caregivers can share and reply to posts from one another, either through direct messaging, comments left on posts, images or videos [35,40]. These forums may include moderation and participation from mental healthcare providers and/or evaluators, but focus mainly on peer-to-peer contact between SSD patients and their immediate social support circle.

*Prompt-Based Intervention (PBI)* is defined in this study as any intervention which provides reminders or alerts that aim to notify the SSD patient of an upcoming goal, commitment or appointment [31,32]. Additionally, these can serve to assure the patient, or encourage self-monitoring through scheduled questionnaires and surveys [20]. The all-encompassing characteristic of these interventions is that they act by pushing notifications in real-time to the patient through their mobile device, instead of delivering a treatment that is accessed by the patient on their own. These can include medication time, scheduled appointments with mental healthcare and social gatherings as well as questions about symptoms and early signs of relapse. They can be delivered through mobile apps or via SMS messaging.

#### 3.2.2. Other Interventions

There are other intervention types, like Virtual Reality Therapy, that can theoretically be delivered remotely, with the increasing accessibility to these devices [47]. However, at the time of this review, and to our knowledge, there have been no attempts at delivering these treatments in a remote fashion, and so none have been eligible for inclusion.

Additionally, digital phenotyping, defined as a moment-by-moment quantification of the patient’s status using data from smartphones and other personal digital devices (i.e., GPS, accelerometer, light sensors, phone usage, speech monitor) [48], can be used in the future to further personalize the delivery of the above-mentioned interventions, through real-time monitoring. However, to our knowledge, due to its recency, there is yet to be a study to evaluate the application of these measurements in any intervention designed to improve SSD outcomes, and so none have been included for review.

#### 3.2.3. Overlap of Intervention Types

As previously stated, these treatment modalities are not mutually exclusive. Rather, it is relatively rare for an intervention to be exclusively based around one of these four types, as many seek to combine these modalities for their potential additive benefit in targeting a certain aspect of SSD.

As such, of the 26 included studies, 22 had a WBT component, nine had WBP, five had OPS and five had PBI. Twelve were exclusively dedicated to WBT, three exclusively to WBP, one exclusively to PBI and none exclusively dedicated to OPS. One intervention gathered all four components of treatment [32].

Figure 3 depicts the current landscape of interventions, as defined by these four modalities, and presents their overlap across the included studies.

The characteristics of all included studies are depicted in Table 1 and all abbreviations used are specified in Figure 4.

**Table 1 ijerph-22-01024-t001:** Characteristics of the included studies. Asterisks (*) refer to statistically significant changes found in a given outcome measure.

Study	Study Aim Origin	Outcome (Measures)	Study Design	Sample Size Attrition	Age (Mean) Gender	Duration/Follow-Up	Intervention
Online Social Cognition Training in Schizophrenia: A Double-Blind, Randomized, Controlled Multi-Site Clinical Trial Nahum et al., 2021 [27]	To compare the efficacy of SocialVille training to an active control (computer games) USA	Social Cognition * (Composite—ER40, PROID, PFMT, MSCEIT-ME, EA) Functional Capacity (UPSA-2) Symptom severity (PANSS) Functioning *(VRFCAT, GFS, SFS *, QLS, SLOF) Motivation (TEPS, BIS/BAS) Facial affect perception * (MFT *) Social perception (TASIT) Theory of Mind (FPRT) Source Memory (SMT), Attributional style (AIHQ)	Parallel-Armed Double-Blind Randomized Controlled Multi-Site Clinical Trial	Total: 108 Intervention: 55 Control: 53 Attrition: 27%	Intervention 42.5 years 53% male Control 43.27 years 49% male	16 weeks	SocialVille Web-Based Therapy An online intervention targeting Social Cognition abilities using 27 individualized SC exercises derived from cognitive training and neuroplasticity principles.
The Effects of Remote Cognitive Training Combined With a Mobile App Intervention on Psychosis: Double-Blind Randomized Controlled Trial Fisher et al., 2023 [35]	To assess the efficacy of 30 h of web-based targeted cognitive and social cognitive training combined with the PRIME app as compared to 30 h of a computer games control condition plus PRIME, in the improvement of deficits in cognition and motivation in people with a psychosis spectrum disorder USA	Cognition * (PCNB *) Motivation * (MAPS *, MSQ *, TEPS *, BIS/BAS) Symptom Severity * (QSANS/QSAPS *) Depression * (BDI *, UCLA-LS) Functioning (QLS *, RFS *)	Parallel-Armed Double-Blind Randomized Controlled Clinical Trial	Total: 100 Intervention: 50 Control: 50 Attrition: 22%	Total 33.77 years 58% male	6 months	PRIME Web-Based Therapy Online Peer Support Auditory Training Module Web-Based Therapy A set of computerized exercises designed to improve the speed and accuracy of auditory information processing while engaging neuromodulatory systems involved in attention and reward Social Cognition Training Module (SocialVille) Web-Based Therapy
A Smartphone App to Promote Patient Activation and Support Shared Decision-making in People with a Diagnosis of Schizophrenia in Outpatient Treatment Settings (Momentum Trial): Randomized Controlled Assessor-Blinded Trial Vitger et al., 2022 [44]	To investigate the effect of a digital tool (Momentum) to support patient activation and SDM as compared to treatment as usual, in people diagnosed with schizophrenia Denmark	Self-reported level of activation * (CHAI-MH *) Self-efficacy (GSE) Hope (ASHS) Working alliance (WAI-S) Satisfaction (CSQ) Preparedness for treatment * (PrepDM *, PEPPI *) Symptom severity (SAPS, SANS) Functioning (GAF, PSP) Number/length of hospital admissions Adherence to appointments	Parallel-Armed Assessor-Blind Randomized Controlled Clinical Trial	Total: 194 Intervention: 96 Control: 98 Attrition: 8.2%	Total 43.1 years 33.5% male 61.9% female 4.6% non-binary	6 months	Momentum Web-Based Therapy Web-Based Psycho-Education A digital shared decision-making tool for the process of cocreation among patients, providers and researchers, with preparation for treatment consultation as the main function.
Web-based psychoeducational intervention for persons with schizophrenia and their supporters: one-year outcomes Rotondi et al., 2010 [37]	To examine the use of a uniquely designed website and home computers to deliver online multifamily psycho-educational therapy to persons with schizophrenia and their informal supports (family and friends). USA	Symptom Severity * (SAPS *, SANS) Knowledge About Disease * (KAS *) Hospitalization (nature, timing, duration, aftermath)	Parallel-Armed Unblinded Randomized Controlled Clinical Trial	Total: 31 Intervention: 16 Control: 15 Attrition: 3%	Total 37.5 years 79% male	1 year	SOAR Web-Based Therapy Web-Based Psycho-Education Online Peer Support A psycho-educational website that provides online group therapy (through forums for both schizophrenia patients and their relatives) as well as psycho-education through direct contact with mental health professionals and a library of educational reading materials.
Single-Session Mobile-Augmented Intervention in Serious Mental Illness: A Three-Arm Randomized Controlled Trial Depp et al., 2019 [34]	To assess the effects of two single-session in-person interventions—CBT2go or Self-Monitoring (SM)—when augmented by mobile interaction with a web-based program (MOBIT) on self-management targets and adaptive beliefs and behaviors of patients with serious mental illness as compared with treatment as usual. USA	Symptom severity * (BPRS-24 *) Functioning * (SLOF *) Defeatist performance beliefs (DPAS)	Three-Armed Assessor-Blind Randomized Controlled Clinical Trial	Total (schizophrenia): 229 (172) Intervention 1 CBT2go: 77 (55) Intervention 2 SM: 69 (54) Control: 83 (63) Attrition: CBT2go—9% SM—15.3%	Intervention 1 51.2 years 54.2% male Intervention 2 49.4 years 47.7% male Control 48.1 years 50.8% male	6 months	CBT2go A single 90-min in-person CBT-based session containing patient-selected modules that aim to define and reshape patients’ self-management strategies and behaviors. SM (Self-Monitoring) A single 90-min in-person psycho-education-based session about the diagnosis, causes, symptoms and treatments for mental illness, and the importance of self-monitoring symptoms. MOBIT Web-Based Therapy Web-Based Psycho-Education Mobile Online Behavioral Intervention Technology, a web-based program delivering interactive surveys to the mobile device containing personalized elements from the individual session.
Randomized controlled trial of an internet cognitive behavioral skills-based program for auditory hallucinations in persons with psychosis Gottlieb et al., 2017 [8]	To assess the efficacy of a Web-based CBTp skills program (Coping with Voices) in improving psychosis symptoms in persons with schizophrenia, as compared to usual care. USA	Hallucination Severity * (BPRS-AH item *, PSYRATS-Auditory Hallucination Subscale *) Symptom severity * (BPRS-24 *, PSYRATS Delusion Subscale *, BAVQ-R *, PS, BDI, BAI) Insight (BCIS) Functioning * (SLOF *)	Parallel-Armed Assessor-Blind Randomized Controlled Clinical Trial	Total: 37 Intervention: 19 Control: 18 Attrition: 19%	Intervention 43.79 years 47.6% male Control 40.28 years 77.8% male	3 months	Coping With Voices (CWV) Web-Based Therapy A 10-module, computerized CBTp intervention for auditory hallucinations that can be completed at the patient’s own pace.
Digitally supported CBT to reduce paranoia and improve reasoning for people with schizophrenia-spectrum psychosis: the SlowMo RCT Garety et al., 2021 [45]	To examine the effectiveness of SlowMo therapy in reducing paranoia and in improving reasoning, quality of life and well-being in people with schizophrenia spectrum psychosis as compared to treatment as usual. UK	Paranoia * (GPTS *, PSYRATS *, SAPS *) Reasoning * (MADS *, EEI *, JTC-BG, FaST-Q) Well-being * (WEMWBS *) Quality of life* (MANSA *) Schemas * (BCSS *) Service use (CSRI) Worry * (PSQW *)	Parallel-Armed Assessor-blind Randomized Controlled Clinical Trial	Total: 362 Intervention: 181 Control: 181 Attrition: 20%	Total 42.6 years 69.8% male	6 months	SlowMo Web-Based Therapy A program consisting of eight individual 60–90-min face-to-face sessions delivered within a 12-week time frame and assisted by the SlowMo web app with interactive features including information, animated vignettes, games and personalized content
Internet-based self-help for psychosis: Findings from a randomized controlled trial Westermann et al., 2020 [38]	To test whether an 8-week, CBT-oriented, Internet-based intervention (IBI) for people with psychosis is feasible, effective and safe compared to care as usual in a waitlist condition. Switzerland, Germany	Symptom Severity * (Composite *: PANSS, LSHS *, PC, MINI) Paranoia (PC) Sleep difficulties (ISI) Hallucination Severity (LSHS) Self-esteem * (RSE *) Meta-cognition (BT) Depression (PHQ-9) Mindfulness * (MAAS *) Worrying (PSWQ-A) Social Competence * (ICQ *) Motivational incongruence (IQ-B) Quality of Life * (WHOQOL *) Internalized Stigma (ISMI) Satisfaction (ZUF-8) Negative experiences/effects (QueSPI)	Parallel-Armed Assessor-blind Randomized Controlled Clinical Trial	Total: 101 Intervention: 50 Control: 51 Attrition: 14%	Total 40.0 years 42% male	8 weeks	iCBTp Program Web-Based Therapy Web-Based Psycho-Education Self-Help program organized in 9 modules addressing single symptoms. One module on relapse prevention and 1 module introducing the program, each containing texts, illustrations, explanatory videos and interactive worksheets. The order of completion of the 9 modules was decided by the patient. A supplemental smartphone app was provided.
Mindfulness and relaxation treatment reduce depressive symptoms in individuals with psychosis Moritz et al., 2015 [26]	To examine the effectiveness of a self-help mindfulness intervention (manual with audio files) in patients with psychosis as compared to a manual on progressive muscle relaxation (PMR) Germany	Symptom Severity * (POD composite: PC, CES-D *, OCI-R *)	Parallel-Armed Assessor-Blind Randomized Controlled Clinical Trial	Total: 90 Intervention: 38 Control: 52 Attrition: 29%	Intervention 38.11 years 42.1% male Control 37.46 years 42.3% male	6 weeks	Mindfulness Manual Web-Based Therapy 15-page manual on mindfulness exercises accompanied by audio files and delivered over the Internet.
Effects of online intervention for depression on mood and positive symptoms in schizophrenia Moritz et al., 2016 [39]	To examine whether an online intervention for depression can ameliorate depressive symptoms in schizophrenia as compared to treatment as usual in a waitlist control condition. Germany	Depression (Composite: CES-D *, PC) Symptom Severity (PHQ-9, PC, PANSS)	Parallel-Armed Assessor-Blind Randomized Controlled Clinical Trial	Total: 58 Intervention: 31 Control: 27 Attrition: 16%	Intervention 38.19 years 54.8% male Control 43.41 years 37% male	12 weeks	HelpID Web-Based Therapy Prompt-Based Intervention Online CBT-based depression training consisting of an individually tailored set of 12 weekly scheduled modules (out of 17 total modules), including videos, interactive exercises and practice sheets, illustrations, photographs, animations and audios. It also made use of motivational SMS (optional) and email reminders as well as personal feedback.
A web-based tool to support shared decision making for people with a psychotic disorder: randomized controlled trial and process evaluation Van der Kriek et al., 2013 [23]	To assess the efficacy of a Web-based intervention in facilitating shared decision making for people with psychotic disorders as compared to treatment as usual. Netherlands	Patient-perceived involvement (COMRADE) Satisfaction * (CSQ *)	Parallel-Armed Unblinded Randomized Controlled Clinical Trial	Total: 250 Intervention: 124 Control: 126 Attrition: 70.8%	Intervention 37 years 67% male Control 40 years 36% male	6 weeks	WegWeis (Web-based information and decision tool) Web-Based Psycho-Education Web-based information and decision tool consisting of 3 webpages and a home page aimed at supporting patients in acquiring an overview of their needs and appropriate treatment options provided by their mental health care organization
The efficacy of SMS text messages to compensate for the effects of cognitive impairments in schizophrenia Pijnenborg et al., 2010 [29]	To evaluate the efficacy of short message service (SMS) text messages to compensate for the effects of cognitive impairments in the daily life of schizophrenia patients, as compared to treatment as usual in a waitlist control condition Netherlands	%Goals achieved * (Appointments *, Medication, Training Programme, Activities and inhibition) Functioning (SFS) Symptom Severity (PANSS) Self-esteem (RSE)	Parallel-Armed Unblinded Un-randomized Waitlist-Controlled Clinical Trial	Total: 62 Intervention: 33 Control: 29 Attrition: 24.2%	Total 28.8. years 79% male	7 weeks	SMS text messages Prompt-Based Intervention Patient-set scheduled SMS messages designed to motivate the completion of self-identified goals. Two SMS messages were sent around each goal (1 h before and 10 min before).
A web-based program to empower patients who have schizophrenia to discuss quality of care with mental health providers Steinwachs et al., 2011 [43]	To evaluate a Web-based tool helping patients with schizophrenia communicate with clinicians about evidence-based treatments, as compared to treatment as usual complimented with an educational video USA	Patient Activation * (RIAS) Duration of visit (minutes) * Number of statements per visit * Clinician verbal dominance * Patient-centeredness ratio *	Parallel-Armed Single-Blind Randomized Controlled Clinical Trial	Total: 50 Intervention: 24 Control: 26 Attrition: N/A	Intervention 49 years 63% male Control 50 years 69% male	N/A (Single Sesssion)	YourSchizophreniaCare Web-Based Psycho-Education Web-based tool to help patients with schizophrenia communicate with clinicians about evidence-based treatments, featuring questions aimed at providing individualized feedback recommendations and complemented by 30 s video clips of actors simulating a patient discussing treatment concerns.
The effect of online group-based acceptance and commitment therapy on psychotic symptoms and functioning levels of individuals with early psychosis Özer et al., 2024 [46]	To examine the effect of online group-based Acceptance and Commitment Therapy on early psychosis patients’ symptoms and functionality levels, as compared to treatment as usual. Turkey	Symptom severity * (PANSS *) Functioning * (SFAS *)	Parallel-Armed Assessor Blind Randomized Controlled Clinical Trial	Total: 53 Intervention: 26 Control: 27 Attrition Intervention: 0% Control: 14.8%	Intervention 23.26 years 76.9% male Control 23.55 years 70.4% male	3 months	Eight-session ACT Web-Based Therapy A program based on online ACT-based 60–90 min bi-weekly group therapy sessions, targeting components of psychological flexibility found at the basis of ACT.
Engagement, clinical outcomes and therapeutic process in online mindfulness for psychosis groups delivered in routine care Ellet et al., 2022 [30]	To examine engagement, clinical outcomes, participant experience and therapeutic process of online delivery of therapy groups in routine clinical practice. UK	Depression * (PHQ-9 *) Anxiety * (GAD-7 *) Symptom Severity * (BAV-Q *)	Single-Armed Pre–Post Clinical Trial	Total: 21 Attrition: 19%	Total 41.73 years 64.7% male	3 months	Online Group PCBT Web-Based Therapy Person-Based Cognitive Therapy (Mindfulness-based Intervention combined with CBT principles) delivered through 12 90-min online mindfulness group sessions.
The Horyzons project: a randomized controlled trial of a novel online social therapy to maintain treatment effects from specialist first-episode psychosis services Alvarez-Jimenez et al., 2021 [42]	To determine the effectiveness of a new digital intervention (Horyzons) in patients with first-episode psychosis, after 2 years, as compared to treatment as usual. Australia	Social Functioning (PSP) Symptom Severity * (PANSS *) Hospitalization Visit to emergency services * Vocational/Educational Recovery * (work/college) Depression (CDSS) Loneliness (UCLA-LS) Social Support (MOS-SSS) Self-esteem (SERS-SF) Self-efficacy (MHCS) Satisfaction with Life (SWLS) Quality of Life (AQoL-8D)	Parallel-Armed Assessor Blind Randomized Controlled Clinical Trial	Total: 170 Intervention: 86 Control: 84 Attrition: 26.7%	Total 20.91 years 52.9% male	18 months	Horyzons Web-Based Therapy Web-Based Psycho-Education Online Peer Support A digital platform merging peer-to-peer social networking (“café” forum), psycho-education and therapeutic interventions through a number of online “pathways” addressing distinct strategies and aspects of the illness.
Supplementing Intensive Targeted Computerized Cognitive Training with Social Cognitive Exercises for People with Schizophrenia: An Interim Report Fisher et al., 2017 [6]	To investigate the effects of supplementing computerized neurocognitive training with social cognitive exercises, in schizophrenia patients, as compared to neurocognitive training alone. USA	Neurocognition * (MCCB, HVLT-R, BVMT-R) Social Cognition * (MSCEIT, PROID *, FPRT *) Motivation * (TEPS *) Symptom Severity (PANSS) Functional Capacity * (UPSA-Brief *) Quality of Life (QLS)	Parallel-Armed Double Blind Randomized Controlled Clinical Trial	Total: 111 Intervention: 57 Control: 54 Attrition: 37%	Intervention 44.08 years 77.2% male Control 42.37 years 64.8% male	6 months (interim report)	SocialVille + TCT Web-Based Therapy An online intervention targeting social cognition abilities using 27 individualized SC exercises derived from cognitive training and neuroplasticity principles, complemented in this study with Targeted Cognitive Training (TCT).
Six-month durability of targeted cognitive training supplemented with social cognition exercises in schizophrenia Miley et al., 2020 [21]	To investigate the long-term effects, 6 months after supplementing computerized neurocognitive training with social cognitive exercises, in schizophrenia patients, as compared to neurocognitive training alone. USA	Neurocognition * (MCCB, HVLT-R, BVMT-R) Social Cognition * (MSCEIT *, PROID *, FPRT *) Motivation * (TEPS *) Symptom Severity * (PANSS *) Functional Capacity * (UPSA-Brief *) Quality of Life (QLS)	Parallel-Armed Double Blind Randomized Controlled Clinical Trial	Total: 34 Intervention: 18 Control: 16 Attrition: 48%	Not specified	12 months 6 months after interim report listed above	SocialVille + TCT Web-Based Therapy An online intervention targeting Social Cognition abilities using 27 individualized SC exercises derived from cognitive training and neuroplasticity principles. Complemented in this study with Targeted Cognitive Training (TCT).
Creating Live Interactions to Mitigate Barriers (CLIMB): a mobile intervention to improve social functioning in people with chronic psychotic disorders. Biagianti et al., 2016 [31]	To investigate the feasibility of delivering 6 weeks of CLIMB to people with Chronic Psychotic Disorders and explore the initial effects on outcomes. USA and Canada	Social Cognition * (PROID *, BLERT *) Quality of Life (SQLS *) Symptom Severity (PANSS)	Single-Armed Pre–Post Clinical Trial	Total: 27 Attrition: 22%	Total 28.1 years 63% male	6 weeks	CLIMB Web-Based Therapy Creating Live Interactions to Mitigate Barriers, an intervention comprised of 2 treatment components: a computerized social cognition training (SCT) program—SocialVille—and optimized remote group therapy (ORGT) consisting of weekly group teletherapy through group texting.
Randomized clinical trial with e-MotionalTraining^®^ 1.0 for social cognition rehabilitation in schizophrenia Maroño Souto et al., 2018 [41]	To test the efficacy of an online self-training program, e-Motional Training^®^, on social cognition in schizophrenia patients as compared treatment as usual. Spain	Symptom Severity * (PANSS) * Cognitive Ability (K-BIT) Social Cognition * (E60FT *, HT *, FPRT, FHSS, MASC *, AIHQ) Emotional Intelligence (MSCEIT) Social Functioning (SFS)	Parallel-Armed Assessor-Blind Randomized Controlled Multicenter Clinical Trial	Total: 60 Intervention: 30 Control: 30 Attrition: 0%	Total 39.17 years 78.3% male	12 weeks	e-Motional Training^®^ Web-Based Therapy A website-delivered online self-training program in social cognition composed of 12 1 h sessions containing tutorials and minigames as well as an animated short film.
Feasibility and outcomes of a multi-function mobile health approach for the schizophrenia spectrum: App4Independence (A4i) Kidd et al., 2019 [32]	To evaluate the feasibility and outcomes of a novel mHealth strategy for schizophrenia and other psychoses called App4independence (A4i). Canada	Symptom Severity * (BSI *) Engagement in Recovery (PROM) Treatment Adherence * (BARS *)	Single-Armed Pre–Post Clinical Trial	Total: 38 Attrition: 0%	Total 31.4 years 71% male	1 month	App4Independence (A4i) Web-Based Therapy Web-Based Psycho-Education Online Peer Support Prompt-Based Intervention A multi-feature self-management app that includes (i) daily prompts for wellness and goal attainment, (ii) evidence-informed content that makes suggestions and provides relevant information and strategies, (iii) a peer–peer engagement platform and (iv) a toolkit and voice detector for offline use.
Feasibility, acceptability, and preliminary efficacy of a smartphone intervention for schizophrenia Ben-Zeev et al., 2014 [20]	To assess the feasibility, acceptability and preliminary efficacy of FOCUSA, a user-centered smartphone system for schizophrenia patients. USA	Symptom Severity * (PANSS *) Depression * (BDI-II *) Sleep Difficulties (ISI) Treatment Adherence (BMQ)	Single-Armed Pre–Post Clinical Trial	Total: 33 Attrition: 3%	Total 45.9 years 61% male	1 month	FOCUS Web-Based Therapy Web-Based Psycho-Education Prompt-Based Intervention Automated real-time/real-place illness management system, comprising 3 apps and a web-based dashboard. It includes brief interactive exchanges accompanied by illustrative images, photographs, cartoons and reminder buttons that can serve as useful information, therapeutic strategy suggestions or reminders.
Improving social cognition in schizophrenia: A pilot intervention combining computerized social cognition training with cognitive remediation Lindenmayer et al., 2012 [25]	To examine whether cognitive remediation (CR; COGPACK) combined with MRIGE, a computerized Emotion Perception intervention (Mind Reading: Interactive Guide to Emotions), improves emotion perception in schizophrenia patients, as compared with CR alone. USA	Social Cognition * (FEIT *, FEDT *, MSCEIT *) Social Functioning * (PSP *) Neurocognition * (MCCB-MATRICS *) Symptom Severity (PANSS)	Parallel Arm Unblinded Randomized Controlled Clinical Trial	Total: 59 Intervention: 32 Control: 27 Attrition: 13%	Total 43.8 years 82% male	3 months	COGPACK CR + MRIGE Web-Based Therapy Computerized Cognitive Remediation Therapy through a computerized, commercially available CR program, combined with MRIGE (Mind Reading: Interactive Guide to Emotions), an interactive computerized program practicing the recognition of emotions and mental states, accessible in library form, as a game or as a learning lesson.
A new computerized cognitive and social cognition training specifically designed for patients with schizophrenia/schizoaffective disorder in early stages of illness: A pilot study. Fernandez-Gonzalo et al., 2015 [36]	To assess the efficacy of a new computerized cognitive and social cognition program, NPT-MH, for patients with schizophrenia/schizoaffective disorder with recent diagnosis, as compared to non-specific computer training. Spain	Neurocognition * (WAIS-III, WmS-III, CPT-II, RAVLT, TMT A, TMT B*, SWCT, VFT, TOL) Social Cognition * (PoFA *, FHSS, HT, RMET, IPSAQ) Symptom Severity (PANSS) Anxiety (STI-trait) Functioning (SFS) Quality of Life (QoLI)	Parallel-Armed Double-Blind Randomized Controlled Clinical Trial	Total: 53 Intervention: 28 Control: 25 Atrition: 24.5%	Intervention 30.9 years 60.7% male Control 30.02 years 68% male	4 months	NPT-MH Web-Based Therapy NeuroPersonalTrainer-Mental Health, a new computerized cognitive and social cognition program, comprising 2 modules: (1) a cognition module which includes attention, memory and executive functions tasks with different levels of complexity, and (2) a social cognition module with 43 tasks based around pictures, stories and videos of common social situations
Efficacy of PRIME, a Mobile App Intervention Designed to Improve Motivation in Young People with Schizophrenia Schlosser et al., 2018 [40]	To assess the efficacy of PRIME, a mobile-based digital health intervention, in improving motivation and quality of life in people with recent-onset Schizophrenia Spectrum Disorders, as compared to treatment as usual in a waitlist control condition. USA	Motivation * (TT *) Defeatist beliefs * (MAP-SR *) Depression * (BDI-II *) Functioning (RFS) Quality of Life (QLS-A) Self-efficacy * (R-SES *) Symptom Severity (PANSS)	Parallel-Armed Assessor -Blind Randomized Controlled Clinical Trial	Total: 43 Intervention: 22 Control: 21 Attrition: 23 %	Intervention 24.32 years 60% male Control 23, 79 years 65% male	3 months	PRIME Web-Based Therapy Online Peer Support Personalized real-time intervention for motivational enhancement, a mobile-based digital health intervention designed to improve motivation and quality of life, through completing sequential, patient-set challenges and sharing them with the PRIME community.
Mobile Assessment and Treatment for Schizophrenia (MATS): a pilot trial of an interactive text-messaging intervention for medication adherence, socialization, and auditory hallucinations Granholm et al., 2012 [33]	To assess the efficacy of ambulatory monitoring methods and cognitive behavioral therapy interventions in improving medication adherence, socialization and auditory hallucinations in schizophrenia patients. USA	Daily assessment question for Treatment adherence * Hallucination severity * Socialization * Symptom Severity (PANSS) Depression (BDI-II) Functioning (ILSS) Cognition (ANART)	Single-Armed Pre–Post Clinical Trial	Total: 55 Atrtition: 24%	Not specified	3 months	MATS Web-Based Therapy Prompt-Based Intervention Mobile Assessment and Treatment for Schizophrenia, ambulatory monitoring methods and CBT-based interventions to assess and improve outcomes in consumers with schizophrenia through mobile phone text messaging targeting 1 of the 3 intervention domains: medication adherence, socialization or auditory hallucinations.

**Figure 4 ijerph-22-01024-f004:**
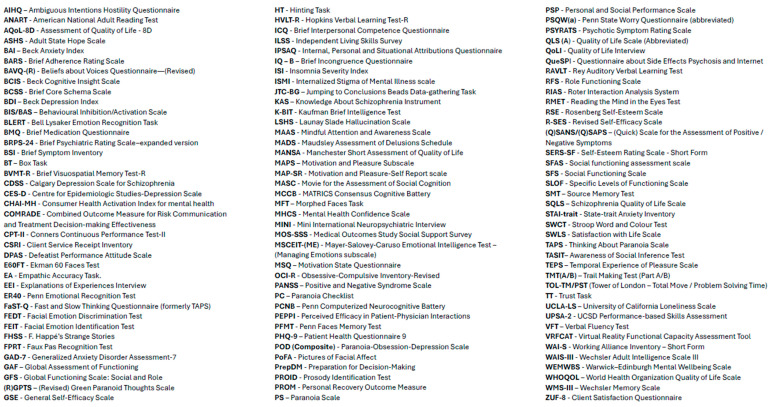
Outcome measure abbreviations.

### 3.3. Narrative Analysis

For this review, we will present the narrative analysis of the included studies by grouping the many outcomes they measure. This was completed in an attempt to shed light on current research goals, as well as possibly indicate which aspects of SSDs are more effectively targeted by these interventions. We also gathered all tools of measurement used for each outcome across all studies, to obtain an understanding of how uniform and homogenous the current research on a given SSD outcome is.

### 3.4. Outcomes

After detailed reading of the presented articles, we selected the following outcomes as the most relevant and commonly appraised: (1) symptom severity (which includes positive or negative symptoms of SSDs, excluding those that overlap with anxiety and depression); (2) anxiety and depression (including defeatist beliefs, loneliness, worry and other associated outcomes); (3) functioning (which encompasses social functioning, community functioning, functional capacity, etc.); (4) motivation and self-esteem; (5) mindfulness (and outcomes related to hope and well-being); (6) social cognition (and all its different components, here divided into facial affect perception, emotional intelligence, prosody identification and theory of mind/social cue perception); (7) neurocognition (cognitive ability, meta-cognition, theory of mind, attributional style, etc.); (8) quality of life; (9) involvement in treatment (in all of its aspects such as patient activation, treatment adherence, knowledge about disease, etc.) and finally (10) other less researched outcomes such as reasoning, schemas and relapse prevention (including hospitalization and visitations to emergency services).

#### 3.4.1. Symptom Severity

A total of 24 studies evaluated whether or not online user-led interventions can lead to improvements in psychotic symptom severity, as well as other positive and negative symptomatology. In 10 of these studies (SocialVille [27], Momentum [44], HelpID [39], SMS [29], CLIMB [31], MRIGE + COGPACK CR [25], NPT-MH [36], PRIME [40], MATS [33]), no significant association was found between the intervention and symptoms.

Conversely, 14 studies (PRIME + SocialVille + Auditory Training [35], SOAR [37], CBT2go + MOBIT [34], Coping with Voices [8], SlowMo [45], iCBTp Program [38], Mindfulness Manual [26], 8-session ACT [46], Online Group PCBT [30], HORYZONS [42], SocialVille + TCT [6,21], E-motional Training^®^ [41], App4Independence [32], FOCUS [20]) found some significant improvements. These results are summarized in Table 2.

##### Findings

One intervention designed for improving social cognition (SocialVille) only found significant results in symptomatology when combined with other interventions (PRIME + SocialVille + Auditory Training [35]; SocialVille + TCT [6,21]), and did not lead to significant improvement on its own (SocialVille [27]). On one of these occasions, the combination of SocialVille and another intervention (SocialVille + TCT [6,21]) only saw significant improvements in symptoms (PANSS General and PANSS Total) after 1 year of follow-up [6]. One other intervention using SocialVille as social cognition training in a single-armed pre–post trial (CLIMB [29]), however, found no significant results in symptom severity over its 6-week intervention.

The combination of SocialVille with PRIME and an Auditory Training module produced significant improvements in positive and negative symptoms after 6-month follow-up, which can be, in part, attributable to the PRIME intervention, as the active control group, which was administered PRIME and conventional computer games, also saw a modest improvement in this outcome [35].

One intervention (SOAR [37]) found similar effects on positive symptoms and indicated that there was a tendency for patients with more severe positive symptoms to spend more time, and access the SOAR site more frequently. However, cumulative site usage and the magnitude of reduction in these symptoms were weakly associated and found not to be statistically significant.

One study which included a study population of both schizophrenia patients and patients with bipolar disorder (CBT2go + MOBIT [34]) found a modest yet sustained reduction in global psychopathology, which differed between these two populations. In bipolar disorder, the effects were more pronounced in the active phase of the intervention, yet were greatly reduced at follow-up. Conversely, in schizophrenia patients, the effects were undetectable in the active phase and were later found in the follow-up evaluation.

Another study, whose intervention was focused around improving auditory hallucinations through a 10-module CBT-based intervention (Coping with Voices [8]), found a significant decrease in overall psychotic symptoms, as well as improved Beliefs About Voices scores. Additionally, it found a significant group–age interaction on negative symptoms: on older patients, the intervention demonstrated a stabilizing effect, as the control group’s negative symptoms tended to worsen at follow-up. On the other hand, younger subjects tended to improve in this outcome regardless if they were assigned to the intervention or control group.

A similar finding was reached in a study on a population of younger, first-episode psychosis patients (HORYZONS [42]), where a digital platform delivering WBT, WBP and OPS led to significant treatment-by-time effects on negative symptoms as measured by the PANSS Negative subscale. No such effect was found on the other subscales. This improvement was observed at 12 months of follow-up, and was subsequently lost at 18 months. E-motional Training^®^ [41], a WBT intervention for social cognition, also found significant improvements in the PANSS Negative subscale, when administered over 12 weeks in a population with a mean age of 39.17 years. This contrasts with the results of a single-armed pre–post trial testing a WBT, WBP, PBI intervention (FOCUS [20]) on an older population, with a mean age of 45.9 years, where significant reduction from pretrial to posttrial (4 weeks) was achieved in PANSS total, PANSS positive and PANSS General, but not on PANSS Negative or BMQ.

When testing the effects of complementing face-to-face individual therapy with a web application (SlowMo [45]) on paranoia, significant effects on all measures of psychotic symptoms (GPTS, PSYRATS and SAPS) were achieved. In opposition to this, a purely remote self-help, Internet-based intervention (iCBTp Program [38]) found a significant decrease with a small to medium effect size in a composite symptom measure, of which self-reported hallucinations had the most prevalent improvement. These findings were found to be comparable to a then-recent network meta-analysis of trials with face-to-face therapies in psychosis [49]. Further on paranoia, a mindfulness and relaxation treatment applied remotely through a manual complimented by audio files (Mindfulness Manual [26]) has shown improvement on a general symptom severity composite, but has only shown significance in obsessive–compulsive and depressive symptoms, not paranoia.

In a clinical trial accessing the efficacy of 8-session group ACT on the symptoms and functioning of early psychosis patients (8-session ACT [46]), a significant reduction was achieved in PANSS total, PANSS positive and PANSS negative sub-measures, both post-intervention and after a 3-month follow-up. When compared to face-to-face ACT, this study showed similar efficacy in reducing general psychopathology, including negative symptoms. Another study on online group therapy for SSD, this one based on a combination of Mindfulness Training and CBT (Online Group PCBT [30]), found clinically significant and reliable improvements in Beliefs About Voices after 12 weeks of the intervention.

Finally, a single-armed pre–post trial on an intervention containing all four treatment modalities—WBT, WBP, OPS and PBI—was successful in finding significant improvements psychoticism, depression, phobic anxiety, obsessive–compulsive, paranoid ideation and interpersonal sensitivity, but only depression was found significantly altered after correction (App4Independence [32]).

##### Outcomes Measures

Outcome measures used to evaluate symptoms varied greatly from study to study, and included the following: PANSS (Positive and Negative Syndrome Scale [50]), SANS/SAPS (Scale for the Assessment of Positive/Negative Symptoms [51]), BPRS-24 (Brief Psychiatric Rating Scale–expanded version [52]), PSY-RATS (Psychotic Symptom Rating Scale [53]) BAV-Q (Beliefs About Voices Questionnaire [54]), PS (Paranoia Scale), GPTS (Green Paranoid Thoughts Scale [55]), OCI-R (Obsessive–Compulsive Inventory-Revised [56]), PHQ-9 (Patient Health Questionnaire 9 [57]), BSI (Brief Symptom Inventory [58]), LSHS (Launay Slade Hallucination Scale [59]).

#### 3.4.2. Depression and Anxiety

A total of 12 studies included an evaluation on the effects of an online user-led intervention on depression/anxiety. Six of these studies (CBT2go + MOBIT [34], Coping with Voices [8], iCBTp Program [38], HORYZONS [42], NPT-MH [36], MATS [33]) found no significant improvements in these outcomes; and six studies (PRIME + SocialVille + Auditory Training [35], SlowMo [45] Mindfulness Manual [26], HelpID [39], Online Group PCBT [30], FOCUS [20]) achieved significant results. These results are summarized in Table 3.

##### Findings

PRIME + SocialVille + Auditory Training [35] produced significant improvements in depressive symptoms, as measured by the BDI (Beck Depression Inventory), but no change in loneliness (UCLA Loneliness Scale). FOCUS [20] was also found to be effective at improving depression, as measured by the BDI. Similar effects on depressive symptoms were achieved over 6 weeks by a self-help mindfulness intervention, Mindfulness Manual [26], as measured by a composite which detected a significant reduction in depressive and obsessive–compulsive symptomatology, but not paranoia. A different study using a composite of paranoia and depressive symptoms found the same pattern of improvement when delivering weekly modules of CBT-based WBT (HelpID [39]) over a 12-week period.

Online Group PCBT [30] produced a significant clinically relevant decrease in both anxiety and depression symptoms over 3 months of WBT group sessions, with large effect sizes. Finally, SlowMo [45] was shown to be effective in reducing worry over a 6-month period.

##### Outcome Measures

Outcome measures used to evaluate depression/anxiety included the following: BDI (Beck Depression Index [60]), UCLA-LS (University of California Loneliness Scale [61]), PHQ-9 (Patient Health Questionnaire 9 [57]), CDSS (Calgary Depression Scale for Schizophrenia [62]), MAP-SR (Motivation and Pleasure-Self Report scale [63]), DPAS (Defeatist Performance Attitude Scale [64]), CES-D (Center for Epidemiologic Studies-Depression Scale [65]) for depression and PSQW (Penn State Worry Questionnaire [66]), BAI (Beck Anxiety Index [67]), GAD-7 (Generalized Anxiety Disorder Assessment-7 [68]) and STAI-trait (State-trait Anxiety Inventory [69]) for anxiety.

#### 3.4.3. Functioning

A total of 14 studies included an evaluation on the effects of an online user-led intervention on functioning. Seven of these studies (Momentum [44], SMS [29], HORYZONS [42], E-motional Training^®^ [41], NPT-MH [36], PRIME [40], MATS [33]) found no significant improvements in this outcome; and seven studies (SocialVille [27], PRIME + SocialVille + Auditory Training [35], CBT2go + MOBIT [34], Coping with Voices [8], 8-session ACT [46], SocialVille + TCT [6,21], MRIGE + COGPACK CR [25]) achieved significant results. These results are summarized in Table 4.

##### Findings

SocialVille, a social cognition-focused WBT intervention, has been successful at improving the level of functioning in SSD patients across all studies in which it was administered, be it alone (SocialVille [27]) or in combination with other interventions (PRIME + SocialVille + Auditory Training [35], SocialVille + TCT [6,21]). When administered on its own, it only achieved significant improvement in social functioning, as measured by the SFS (Social Functioning Scale). All other measures of functioning (Global Assessment of Functioning, GFS and Virtual Reality Functional Capacity Assessment Tool, VRFCAT) showed no significant changes. Functional capacity, as measured by UPSA-2 (UCSD Performance-based Skills Assessment), was also unaffected. This contrasts with the findings observed when combining SocialVille with Targeted Cognitive Therapy (SocialVille + TCT [6,21]), which have produced significant improvement in UPSA-brief scores that were maintained over 6-month and 12-month follow-ups. PRIME + SocialVille + Auditory Training [35] has also been shown to improve functioning as measured by the RFS (Role Functioning Scale).

Two other studies (8-session ACT [46], MRIGE + COGPACK CR [25]) have shown efficacy in significantly improving social functioning as measured by the SFAS (Social functioning Assessment Scale) and PSP (Personal and Social Performance Scale), respectively.

Significant group-by-time improvements have also been found in a three-armed randomized trial, with the administration of CBT2go + MOBIT [34], a WBT intervention combining single-session CBT therapy and a mobile app delivering surveys. This experimental arm was favored over both TAU and Self-Monitoring + MOBIT in measurements of community functioning using SLOF (Specific Levels of Functioning Scale). Further, scores in the TAU condition showed a downward trend over the 24-week period, as opposed to the experimental conditions, CBT2go + MOBIT and Self-Monitoring + MOBIT. Despite this, the average estimated change in SLOF was not significant in any of the three conditions. Conversely, Coping with Voices11 found significant improvements in overall SLOF and SLOF Interpersonal Functioning over 3 months of follow-up.

##### Outcome Measures

Outcome measures used to evaluate functioning included the following: VRFCAT (Virtual Reality Functional Capacity Assessment Tool [70]), GAF (Global Assessment of Functioning [71]), SFS (Social Functioning Scale [72]), SLOF (Specific Levels of Functioning Scale [73]), RFS (Role Functioning Scale [74]), PSP (Personal and Social Performance Scale [75]), UPSA-Brief (UCSD Performance-based Skills Assessment [76]), ILSS (Independent Living Skills Survey [77]).

#### 3.4.4. Motivation

A total of seven studies included an evaluation on the effects of an online user-led intervention on motivation. Three of these studies (SocialVille [27], SMS [29], HORYZONS [42]) found no significant improvements in this outcome; and four studies (PRIME + SocialVille + Auditory Training [35], iCBTp Program [38], SocialVille + TCT [6,21], PRIME [40]) achieved significant results. These results are summarized in Table 5.

##### Findings

PRIME, a WBT and OPS intervention specifically targeted towards motivation and quality of life, was shown to be effective in improving motivation in a significant manner, both when administered on its own (PRIME [40]) over 3 months to a population of recent-onset SSD patients, and combined with social cognition training in an adult population of SSD patients over a 6-month period. The former has led to significant change in Trust Task (TT) scores, whereas the ladder, has seen significant results in the MAPS (Motivation and Pleasure Subscale), MSQ (Motivation State Questionnaire) and TEPS (Temporal Experience of Pleasure Scale), but not BIS/BAS (Behavioral Inhibition/Activation Scale). In studies where the same social cognition training is administered on its own without PRIME, results have either remained significant over 6 and 12 months in measures such as TEPS (SocialVille + TCT [6,21]), or not shown significance (SocialVille [27]).

Additionally, self-esteem has been significantly positively affected by the iCBTp Program [38], as measured by the RSE (Rosenberg Self Esteem Scale), but not by SMS [29] or HORYZONS [42].

##### Outcome Measures

Outcome measures used to evaluate motivation and self-esteem included the following: TEPS (Temporal Experience of Pleasure Scale [78]), BIS/BAS (Behavioral Inhibition/Activation Scale [79]), MAPS (Motivation and Pleasure Subscale [63]), MSQ (Motivation State Questionnaire [80]), RSE (Rosenberg Self-Esteem Scale [81]), SERS-SF (Self-Esteem Rating Scale—Short Form [82]) and TRUST Task (Timed Reconstruction of Unseen Structures Together [83]).

#### 3.4.5. Mindfulness

A total of three studies included an evaluation on the effects of an online user-led intervention on mindfulness. One of these studies (Momentum [46]) found no significant improvements in this outcome; and two studies (SlowMo [45], iCBTp Program [38]) achieved significant results. These results are summarized in Table 6.

##### Findings

The iCBTp Program [38], a WBP and WBT intervention, produced a significant interaction of time by condition for mindfulness score measured using the MAAS (Mindful Attention and Awareness Scale) when administered to SSD patients, when compared to those in a waitlist condition. Similarly, WBT intervention SlowMo [45] has shown to be effective in improving well-being, measured using the WEMWBS (Warwick–Edinburgh Mental Wellbeing Scale) over a 6-month period. In other studies (Momentum [44]) measuring hope using the ASHS (Adult State Hope Scale), no significant change was observed.

##### Outcome Measures

Outcome measures used to evaluate mindfulness, well-being and hope included the MAAS (Mindful Attention and Awareness Scale [84]), WEMWBS (Warwick–Edinburgh Mental Wellbeing Scale [85]) and ASHS (Adult State Hope Scale [86]).

#### 3.4.6. Social Cognition

A total of seven studies included an evaluation on the effects of an online user-led intervention on social cognition (SocialVille [26], iCBTp Program [41], SocialVille + TCT [6,40] CLIMB [30], E-motional Training^®^ [8], MRIGE + COGPACK CR [24], NPT-MH [38]). All of them achieved significant results. These results are summarized in Table 7.

##### Findings

Various purpose-built interventions for social cognition found success in improving this outcome. SocialVille led to significant change both on its own (SocialVille [27]) and when combined with Targeted Cognitive Training (SocialVille + TCT [6,21]). The former trial (SocialVille [27]) has found significant group × time interaction on a social cognition composite score created from PFMT (Penn Faces Memory Test), PROID (Prosody Identification Test), ER40 (Penn Emotional Recognition Test), MSCEIT (Mayer–Salovey–Caruso Emotional Intelligence Test) and EA (Empathic Accuracy Task) maintained over a 16-week follow-up, as well as a significant increase in facial affect perception, as measured by MFT (Morphed Faces Task) with no significant improvement found in Social Perception as measured by TASIT (Awareness of Social Inference Test). The latter trial (SocialVille + TCT [6,21]) achieved significant results in similar testing, as MSCEIT, PROID and FPRT were all positively affected over a 6-month and 12-month period. Improvements in PROID of a significant magnitude have also been achieved with CLIMB [31], a WBT intervention that also uses SocialVille. It has also seen improvement in BLERT (Bell Lysaker Emotion Recognition Task).

Another social cognition-targeting intervention, E-motional Training^®^ [41], found significant results over a 12-week period in E60FT (Ekman 60 Faces Test), HT (Hinting Task) and MASC (Movie for the Assessment of Social Cognition) with a large size effect, but found no significant changes in AIHQ (Ambiguous Intentions Hostility Questionnaire) other than the Aggression component, FHSS (F.Happé’s Strange Stories), FPRT (Faux Pas Recognition Test) or MSCEIT.

NPT-MH [38], a WBT intervention, was able to improve facial affect perception, as measured by PoFA (Pictures of Facial Affect) scores over a 4-month follow-up, but not by RMET (Reading the Mind in the Eyes Test). Theory of mind, as measured by IPSAQ (Internal, Personal and Situational Attributions Questionnaire), FHSS and HT, was not significantly changed.

The administration of a combination of Computerized Cognitive Remediation Therapy with emotional recognition training, MRIGE + COGPACK CR [25], resulted in significant improvement of facial affect perception, as measured by FEIT (Facial Emotion Identification Test) and FEDT (Facial Emotion Discrimination Test) scores, as well as MSCEIT, maintained over 3 months.

Finally, the iCBTp Program [38] has improved social competence, as measured by the ICQ (Brief Interpersonal Competence Questionnaire) over an 8-week intervention.

##### Outcome Measures

Outcome measures used to evaluate social cognition included the following: MSCEIT (Mayer–Salovey–Caruso Emotional Intelligence Test [87]) and EA (Empathic Accuracy Task [88]) for emotional intelligence; FEIT (Facial Emotion Identification Test [89]), FEDT (Facial Emotion Discrimination Test [89]), E60FT (Ekman 60 Faces Test [90]), MFT (Morphed Faces Task [91]), BLERT (Bell Lysaker Emotion Recognition Task [92]), PFMT (Penn Faces Memory Test [93]), ER40 (Penn Emotional Recognition Test), PoFA (Pictures of Facial Affect [94]) and RMET (Reading the Mind in the Eyes Test [95]) for facial affect perception; PROID (Prosody Identification Test) for prosody identification and FPRT (Faux Pas Recognition Test [96]), MASC (Movie for the Assessment of Social Cognition [97]), TASIT (Awareness of Social Inference Test [98]), FHSS (F. Happé’s Strange Stories [99]), HT (Hinting Task [100]), AIHQ (Ambiguous Intentions Hostility Questionnaire [101]) and IPSAQ (Internal, Personal and Situational Attributions Questionnaire [102]) for social cue perception/theory of mind.

#### 3.4.7. Neurocognition

A total of eight studies included an evaluation on the effects of an online user-led intervention on neurocognition. Four studies (SocialVille [27], iCBTp Program [38], E-motional Training^®^ [41], MATS [32]) found no significant results. Four other studies (PRIME + SocialVille + Auditory Training [35], SocialVille + TCT [6,21], MRIGE + COGPACK CR [25], NPT-MH [36]) achieved significant results. These results are summarized in Table 8.

##### Findings

The SocialVille program has produced improvements is neurocognition when combined with PRIME and Auditory Training [35], as well as Targeted Cognitive Training [6,21], but has seen no significant outcome change when applied by itself (SocialVille [27]).

PRIME + SocialVille + Auditory Training [35] saw significant improvement in the PCNB (Penn Computerized Neurocognitive Battery) measure over its 6-month follow-up and SocialVille + TCT [6,21] produced significant results in MCCB (MATRICS Consensus Cognitive Battery), HVLT-R (Hopkins Verbal Learning Test-R) and BVMT-R (Brief Visuospatial Memory Test-R), showing significant main effects of time in global cognition, attention, speed of processing, verbal learning (d = 0.28), visual learning and problem solving at the 6-month mark, although no group-by-time interactions were found to be significant. At the 12-month follow-up, all of these effects were maintained, and a group-by-time interaction at trend-level significance was found in Verbal Learning and Visual Learning.

A similar combination of social cognition and neurocognition targeting (MRIGE + COGPACK CR [25]) found the same significant improvement in MCCB over a 3-month follow-up. Finally, NPT-MH [36], another WBT intervention targeting these outcomes, found significance in changes in component B of the Trailmaking Test (TMT B), but not in TMT A or WAIS-III (Wechsler Adult Intelligence Scale III), WmS-III (Wechsler Memory Scale), CPT-II (Conners Continuous Performance Test-II), RAVLT (Rey Auditory Verbal Learning Test), SWCT (Stroop Word and Color Test) or VFT (Verbal Fluency Test) or TOL (Tower of London Test).

##### Outcome Measures

Outcome measures used to evaluate social cognition included the following: SMT (Source Memory Test), PCNB (Penn Computerized Neurocognitive Battery [93]), MCCB (MATRICS Consensus Cognitive Battery [103]), HVLT-R (Hopkins Verbal Learning Test-R [104]), BVMT-R (Brief Visuospatial Memory Test-R [105]), K-BIT (Kaufman Brief Intelligence Test [106]), WAIS-III (Wechsler Adult Intelligence Scale III [107]), WmS-III (Wechsler Memory Scale [108]), CPT-II (Conners Continuous Performance Test-II [109]), RAVLT (Rey Auditory Verbal Learning Test [110]), TMT A/B (Trail Making Test Part A/B [111]), SWCT (Stroop Word and Colour Test [112]), VFT (Verbal Fluency Test [113]), TOL (Tower of London [114]), ANART (American National Adult Reading Test [115]) and BT (Box Task [116]).

#### 3.4.8. Quality of Life

A total of seven studies included an evaluation on the effects of an online user-led intervention on quality of life. Four studies (HORYZONS [42], SocialVille + TCT [6,21], NPT-MH [36], PRIME [40]) found no significant results. Three other studies (SlowMo [45], iCBTp Program [38] CLIMB [31]) achieved significant results. These results are summarized in Table 9.

##### Findings

SlowMo [45], a WBT intervention centered around decreasing paranoia in SSD patients, found success in both outcomes, with a significant improvement in MANSA (Manchester Short Assessment of Quality of Life) score over 6 months of follow-up. Another WBT intervention, CLIMB [31], using SocialVille and group texting therapy sessions to improve social cognition and symptom severity, managed to achieve significant changes in quality of life, as measured by SQLS (Schizophrenia Quality of Life Scale) over a 6-week period. Other interventions in social cognition have failed in this regard (SocialVille + TCT [6,21], NPT-MH [36]).

Finally, the iCBTp Program [38], a WBT/WBP intervention aiming to improve SSD symptoms and reduce risk of relapse, led to a significant improvement in quality of life, measured using the WHOQOL (World Health Organization Quality of Life Scale).

##### Outcome Measures

Outcome measures used to evaluate quality of life included the following: MANSA (Manchester Short Assessment of Quality of Life [117]), WHOQoL (World Health Organization Quality of Life Scale [118]), AQoL-8D (Assessment of Quality of Life—8D [119]), QLS (Quality of Life Scale [120]), SQLS (Schizophrenia Quality of Life Scale [121]), QoLI (Quality of Life Interview), SWLS (Satisfaction with Life Scale [122]).

#### 3.4.9. Involvement in Treatment

A total of 10 studies included an evaluation on the effects of an online user-led intervention on treatment involvement. Three studies (WegWeis [23], HORYZONS [42], FOCUS [20]) found no significant results. Seven other studies (Momentum [44], SOAR [37], SMS [29], YourSchizophreniaCare [43], App4Independence [32], PRIME [40], MATS [31]) achieved significant results. These results are summarized in Table 10.

##### Findings

YourSchizophreniaCare [43], an intervention designed to equip SSD patients with better knowledge of how to discuss their symptoms and treatment with clinicians, found a significant improvement over a single session, in duration of visit, number of statements per visit and patient-centeredness ratio, as well as a reduction in clinician verbal dominance, as measured after coding of the consultation transcript using the RIAS system. Another “pure” WBP intervention providing information to SSD patients about treatment options according to their needs, WegWeis [23], found no such significant results, however, and faced an abnormally high 70.8% attrition rate.

Another WBP intervention, Momentum [44], found significant improvement over 6 months in self-reported level of activation as measured by the CHAI-MH (consumer Health Activation Index for mental health) as well as preparedness for treatment as measured in PrepDM (Preparation for Decision-Making) and PEPPI (Perceived Efficacy in Patient-Physician Interactions) scores. The same significance was not reached in working alliance, self-efficacy or adherence to appointments.

When it comes to WBP interventions that include OPS elements, SOAR [37], which combines a psycho-educational website with an online forum for schizophrenia patients and their relatives, has produced significant improvement in knowledge about disease, as measured by KAS (Knowledge About Schizophrenia Instrument) over a 1-year follow-up. PRIME [40], an intervention set on improving motivation in recent-onset psychosis patients through the completion of patient-set challenges, was successful in improving self-efficacy, as measured by the R-SES (Revised Self-Efficacy Scale) over a period of 3 months.

Conversely, HORYZONS [42], aimed at providing FEP patients with appropriate strategies to deal with various burdens of illness, has shown no significant increase in self-efficacy, as measured by the MHCS (Mental Health Confidence Scale).

As for interventions with a PBI element, SMS [29], which aimed to motivate the completion of patient-set goals through daily delivery of related text messages over 7 weeks, produced a significant increase in percentage of goals achieved. The overall mean success percentage was 47% during baseline, and increased to 62% during the intervention. These results were not maintained after withdrawal of the text messages, as the percentage of goals completed dropped to 40% at follow-up. When subdivided into four categories—appointments, medication, training program and activities/inhibition—only the first one found significance and it too did not withstand after intervention withdrawal. Additionally, results suggested that the most severely impaired patients will benefit the most from prompting.

Another PBI intervention, MATS [33], which was assessed in a single-armed pre–post trial, used text messages to target three aspects of SSD: medication adherence, socialization and auditory hallucinations, through daily assessment questions. It found a significant improvement in treatment adherence and reporting, as well as in beliefs about medication helpfulness. This effect was most pronounced in patients living independently, initially less likely to adhere to medication, as they caught up and surpassed those in assisted living situations. When it comes to socialization, it led to improved chances of social interaction, as well as a more positive view of such encounters. Finally, on hallucinations, it found a reduction in the probability of bothersome symptoms, as well as a more balanced view of them.

Conversely, another single-armed pre–post trial on FOCUS [20], which administered prompts featuring images, surveys and information to schizophrenia patients, found no such results across its 1-month runtime, as measured using the BMQ (Brief Medication Questionnaire).

Finally, an intervention which combined all four treatment modalities (WBT, WBP, PBI and OPS) through a multi-feature self-management app, App4Independence [30], was found in a single-armed pre–post trial to increase treatment adherence in a significant manner, as measured by the BARS (Brief Adherence Rating Scale). It found no significance, however, in engagement in recovery measured by the PROM (Personal Recovery Outcome Measure) after 1 month of intervention.

##### Outcome Measures

Outcome measures used to evaluate patient involvement included the following: CHAI-MH (Consumer Health Activation Index for mental health [123]), GSE (General Self-Efficacy Scale [124]), WAI-S (Working Alliance Inventory—Short Form [125]), PrepDM (Preparation for Decision-Making [126]), PEPPI (Perceived Efficacy in Patient-Physician Interactions [127]), KAS (Knowledge About Schizophrenia Instrument [128]), COMRADE (Combined Outcome Measure for Risk Communication and Treatment Decision-making Effectiveness [129]), %Goals Achieved, RIAS (Roter Interaction Analysis System [130]), PROM (Personal Recovery Outcome Measure [131]), BARS (Brief Adherence Rating Scale [132]), BMQ (Brief Medication Questionnaire [133]), R-SES (Revised Self-Efficacy Scale [134]).

#### 3.4.10. Other Outcomes

SlowMo [45], a WBT intervention successfully targeting paranoia, found significant improvement in reasoning over a 6-month period, as measured by the MADS (Maudsley Assessment of Delusions Schedule [135]) and EEI (Explanations of Experiences Interview). It found, however, no improvements in the JTC-BG (Jumping to Conclusions Beads Data-gathering Task [136]) and FaST-Q (Fast and Slow Thinking Questionnaire [45]). Additionally, it succeeded in improving schemas as measured in the BCSS (Brief Core Schema Scale [137]).

Finally, in relapse prevention, a randomized clinical trial studying HORYZONS [42] in FEP patients over 18 months found a hospitalization rate half of that of TAU in the experimental group, although this change was not found to be significant. Visits to emergency services, however, were significantly reduced by the intervention to half of those observed in the control condition. Vocational and/or educational outcomes were also increased, as participants in the active condition had 5.5 times greater chances at finding employment and/or college enrolment. SOAR [37] and Momentum [44], however, were unsuccessful in reducing hospitalization rate or length in any significant way.

## 4. Discussion

This study evaluated 26 studies testing the outcomes of online, user-led interventions, to which we attributed one or more of four possible treatment modalities, in order to analyze their most commonly measured outcomes one by one in a narrative fashion. The main findings of this analysis are as follows:


*Online User-Led Interventions are effective at treating SSD Symptoms.*


Findings on the effects of online user-led interventions on symptoms in SSD suggest the following: (1) the combination of different interventions may have a cumulative effect, combining improvements seen in other outcomes of the illness, with improvements in symptomatology; (2) there may be a time-delayed effect of some interventions in schizophrenia patients, with clinically observable results in symptoms being achieved only after an extended follow-up time, as opposed to other SMI populations, where the effect is more immediate; (3) there may be a tendency for greater improvements in negative symptoms in younger populations, as several studies involving younger populations, earlier into the course of disease, have pointed out and (4) online group therapy, be it ACT, CBT, Mindfulness Training or a combination of the these modalities, has been found to be comparable to face-to-face in-office therapy in improving SSD symptoms, particularly auditory hallucinations and paranoia.


*Online User-Led Interventions lead to improvements in depression and anxiety.*


There have been found clinically significant improvements in depression and anxiety after the administration of online user-led interventions. It appears that most of the interventions producing significant results are based around WBT. This effect could be related to the similarly significant improvements found in negative symptoms, which point to a possible use case for these interventions, as these are typically hard-to-reach treatment goals.


*Social functioning appears to benefit from WBT interventions targeting social cognition and Motivation. Motivation improvements can be achieved through direct targeting or through improvement in social cognition.*


Among all functioning measures obtained, social function seemed to be the most commonly affected in a significant and positive way by the use of online user-led interventions. Those targeting social cognition (SocialVille [27], SocialVille + TCT [6,21]) and motivation combined with social cognition training (PRIME + SocialVille + Auditory Training [35]) but not motivation alone (PRIME [40]). In turn, motivational improvements can be obtained through motivation-targeting interventions both on their own (PRIME [40]) and combined with social cognition (PRIME + SocialVille + Auditory Training [35]), as well as through social cognition exercises alone (SocialVille + TCT [6,21]).


*Mindfulness seems to be scarcely evaluated, but can benefit from WBT interventions.*



*Social Cognition holds promise as a successfully improved target of SSD treatment through online user-led interventions.*


All included interventions targeting social cognition have achieved significant improvement in at least one facet of social cognition [138]: emotion perception, theory of mind (ToM), and attributional style, here divided into emotional intelligence, facial affect perception, prosody identification and theory of mind/social cue perception. The most successful WBT interventions in improving SC (SocialVille [27], iCBTp Program [38], SocialVille + TCT [6,21], E-motional Training^®^ [41], MRIGE + COGPACK CR [25]) have been administered to populations with a mean age of around 40 years. Further research is needed into the effects of these interventions in younger populations, where tendential improvements in negative symptoms have been documented through similar interventions.

There was a noticeable overlap between studies on WBT interventions targeting social cognition that achieved significant results in both social cognition and functioning (SocialVille [27], SocialVille + TCT [6,21], MRIGE + COGPACK CR [25]), which could point out this outcome as an indirect pathway to improvements in functioning, and possibly other outcomes that are similarly affected by negative symptoms.

Social cognition has proven to lend itself well to a remote approach, as many exercises and materials composing the various reviewed interventions relied on images and videos, which can easily be administered without major involvement of mental health workers. As an adjuvant treatment to the conventional one (pharmacological therapy and psychotherapy), this approach could be the answer to the difficult-to-treat negative symptoms of SSD in a way that is scalable, user-led and non-burdening towards mental healthcare providers.


*Cognition improvements seem to lead to social cognition improvement, but not vice-versa.*


Social cognition interventions do not seem to lead to great improvements in neurocognition on their own (SocialVille [27], E-motional Training^®^ [41]) but simultaneous targeting of overall cognition seems improve to both outcomes (PRIME + SocialVille + Auditory Training [35], SocialVille + TCT [6,21], MRIGE + COGPACK CR [25], NPT-MH [36]). Like social cognition, this outcome can be easily targeted in a remote fashion and so, the association of SC and cognition training could be beneficial in boosting the positive effects of the former.


*Online User-Led interventions can improve quality of life, seemingly through symptom severity reduction, and are more effective in chronic SSD patients.*


Among those interventions that tackle quality of life, those that focus on reducing symptomatology and risk of relapse (SlowMo [45], iCBTp Program [38], CLIMB [31]) tended to yield better results than those focusing on the improvement of social cognition and functioning (SocialVille + TCT [6,21], NPT-MH [41]). Additionally, interventions administered to younger populations of patients with first-episode psychosis (HORYZONS [42] with a mean age of 20.91 years) or recent-onset schizophrenia (PRIME [40] with a mean age of 24.32 years in the intervention group) did not find meaningful improvement in quality of life. It could be inferred that older SSD patients, further along the disease course, stand to benefit more from the added improvements, mostly in hard-to-control negative symptoms, of online user-led interventions, as their conventional pharmacotherapy and psychotherapy may already be optimized, whereas younger patients may benefit from harder-to-achieve improvements, such as those seen in social cognition that these symptom-focused interventions mostly fail to cover.


*WBP interventions with OPS and PBI elements seem to be most effective at improving patient involvement in treatment.*


In opposition to WBT-centered interventions, which seem to be more effective in reducing symptoms and improving cognitive impairment, interventions which increase SSD patient involvement in their treatment seem to lean towards psycho-educational interventions (YourSchizophreniaCare [43], Momentum [44]) as well as those that feature prompting (SMS [29], MATS [33]) and peer-to-peer connection (SOAR [37], PRIME [40]). The former seem to be most effective in increasing medication and appointment adherence, an effect which seems to be amplified in those who live independently and, therefore, stand to benefit more, as they lack assistance in remembering such commitments. The latter interventions seem to ameliorate knowledge of the illness, as well as the notion of self-efficacy, as contact with peers may lead to a more informed and hopeful-about-recovery patient. Purely WBP interventions (YourSchizophreniaCare [43], Momentum [44]) seem to be more effective at empowering and informing the SSD patient in discussing their illness with a clinician, as those included in this study improved self-reported activation and treatment-involvement indicators during consultation, but not appointment or medication adherence.

These outcomes may benefit further from the inclusion of caregivers within the treatment framework. PBI and OPS interventions specifically may be better suited for this integration, as both monitoring of adherence and communication among peers can augment the caregiver’s ability to attend to the patient’s needs as well as their own. WBP can likewise improve knowledge about disease in caregivers. More research on the effects of online user-led interventions specifically on caregivers is needed.

Online, user-led interventions can lead to improvements in relapse prevention, but are scarcely studied and have produced mixed results.

### Limitations and Future Research

One key limitation is the low number of databases searched, limited in part by access to certain databases. This, however, has not prevented the gathering of a complete and varied assortment of clinical trials, as in the last decade, the research on this topic has seen a prominent rise in interest.

The inaccessibility of some research data, as well as the wide array of measurements used on one given outcome, severely dampened our ability to extract homogenous data for meta-analysis.

A lack of quantitative analysis hinders the strength of our conclusions, as they are based solely on a narrative perspective. Although an important step towards a better understanding of how these interventions may be useful, this study leaves many questions about how their effects compare to those of treatment as usual. A future meta-analysis on some of the outcomes presented is sorely needed, as there has been, to date, only one [14], in 2016, which was only able to compare two studies. Eight years later, the reality is much different, as in this study, 26 clinical trials on these interventions were gathered, 24 of which measured the effects on the same outcome (symptoms), and yet we found great difficulty in gathering data pertaining to the same outcome measure. For symptoms alone, we found 11 different tools used, some more general in their appraisal, some more focused on specific aspects of psychopathology. When it comes to social cognition, the number of measurement tools used increases to 19, with even more variety in what is effectively measured.

As a result, even with the significant growth of this research field over the last decade, it is still very premature to point out a definitive use for these interventions. The research, although plentiful, remains scattered and speculative in its direction.

For future research on this topic, we recommend a more standardized measurement, mainly for two of the appraised outcomes: symptom severity and social cognition. We recommend using the PANSS scale for symptoms as it is an established and widely implemented scale for schizophrenia symptoms as well as the most commonly used among the many symptom-rating scales employed in the current research. Additionally, newer scales, such as the Clinical Assessment Interview for Negative Symptoms (CAINS) and Brief Negative Symptom Scale (BNSS), may in the future prove to be better-suited to evaluate the effect of these interventions—whose most sought-after effect is precisely on negative symptoms [55]. These scales, however, likewise neglect important metrics like impact on psychosocial and cognitive factors, and so, it is recommended to combine PANSS and these negative symptom scales with cognitive measurements, especially those aimed towards social cognition. This multidimensional and complex outcome could benefit from an agreed-upon categorization and simplification in order to provide a future review on this topic with ample ground to more reliably study the promising effects that the current research seems to point to. As a suggestion, we recommend dividing the outcome measurement for social cognition into four dimensions: facial affect perception, which can be measured, for example, with PFMT, emotional intelligence, which can be measured by MSCEIT, prosody identification, using PROID and, finally, theory of mind/social cue perception, which can be assessed using FPRT. This logic applies to all other measured outcomes, but was found to be crucial for symptoms and social cognition.

Additionally, we recommend the categorization of intervention types, such as the one presented in this study, so that, as our knowledge of the effects of each type increases, we may direct them towards whichever outcomes they target most effectively. For example, WBT interventions may benefit from predominantly targeting social cognition, whereas PBI, WBP and OPS may be more useful when targeting patient activation and involvement in treatment.

A more homogeneous and outcome-directed approach to the research on online user-led interventions could be the step needed for a future quantitative analysis of their effects, and the start of a new offering in SSD treatment, as we understand more about its link with cognition and, consequently, functioning.

Finally, future reviews may benefit from the inclusion of interventions which implement Artificial Intelligence (AI). For this study, we have excluded all papers purely focused on data collection and digital phenotyping. However, future interventions may benefit from the collection of patient metrics (gathered through wearable devices, smartphones or social media) to customize and deliver treatment more effectively. Although this has already been implemented in interventions for bipolar disorder [11], currently, we have found no studies implementing AI into an intervention for SSD. This is very likely to change, so we recommend the inclusion of “Artificial Intelligence” and related terms in the search query of future reviews.

Whereas in the first decade of this century we have seen proven the feasibility and usability of these interventions, we are now beginning to gather evidence around their efficacy in addressing long sought-after improvements in SSD burden of disease, with increasing scale and robustness. As we delve into the second half of the 2020s, we must ensure this research is focused and directed towards the outcomes most effectively targeted, and that these outcomes are consistently measured in a way that facilitates, rather than hinders, comparison between studies. It is safe to say we have minded the gap left in SSD treatment, even after conventional approaches are optimized. It is now time to conclude which interventions work, and in which ways they can be implemented into clinical practice, in hopes to, eventually, leave no aspect of SSD untreated, and so mend the therapeutic gap of schizophrenia and its related disorders.

## 5. Conclusions

Online user-led interventions for schizophrenia hold potential as a novel modality of treatment, targeting hard-to-treat components of the overall burden of SSDs. Although their effect on symptom severity seems to be modest, they have consistently been found to be successful at tackling both social cognition and functioning as compared to treatment as usual. Intervention type seems to condition which outcome is improved: symptoms and cognition benefit mostly from WBT interventions, whereas PBI, WBP and OPS interventions are more closely associated with improvements in overall involvement with treatment. The combination of intervention types and their effects is uncertain. Future research on this topic is required in order to quantify these effects. A more streamlined and standardized research strategy, both in intervention classification and outcome measurement, is highly recommended in order to amass sufficient, homogeneous data for quantitative analysis.

If confirmed, the promising results found among these studies, especially those focused on social cognition and functioning, may lead to future implementation of adjuvant treatment for SSDs that targets the cognitive decline that has, so far, been scarcely prevented in an effective manner. Furthermore, the administration of this new treatment modality through an online medium, combined with a user-led approach, can greatly benefit patients at a fraction of the cost of conventional face-to-face interventions. Additionally, the flexibility and scalability of the reviewed interventions allow for constant improvements as well as customization of the treatment to each patient, as technologies such as Artificial Intelligence grow more sophisticated and accurate.

The future holds promise for this research field, as new user-led interventions are conceived, tested and even integrated with previously developed ones. The natural next step in this line of research will be gathering robust quantitative evidence around the efficacy of these interventions in order to support and legitimize their prescription, alongside conventional pharmacotherapy and psychotherapy, to treat the full spectrum of SSD burden, and thus, mend the therapeutic gap.

## Figures and Tables

**Figure 1 ijerph-22-01024-f001:**
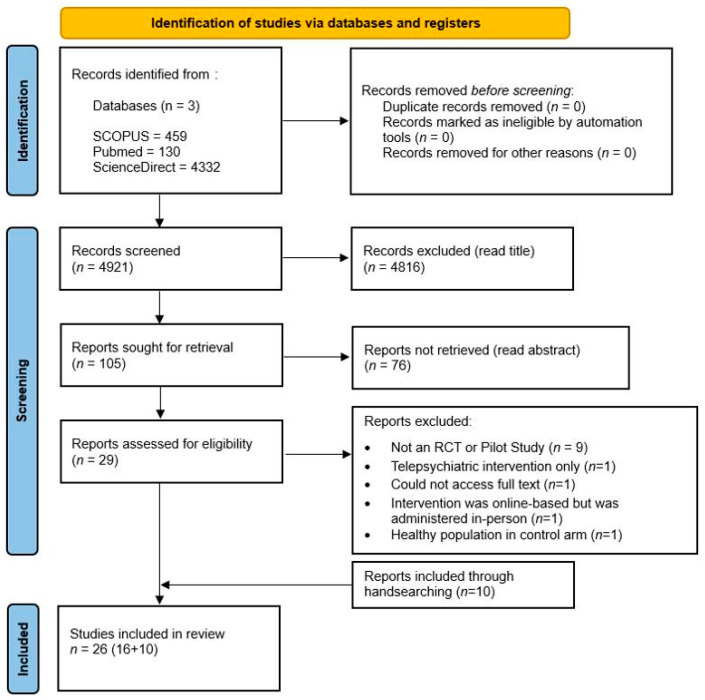
PRISMA flowchart for study selection.

**Figure 3 ijerph-22-01024-f003:**
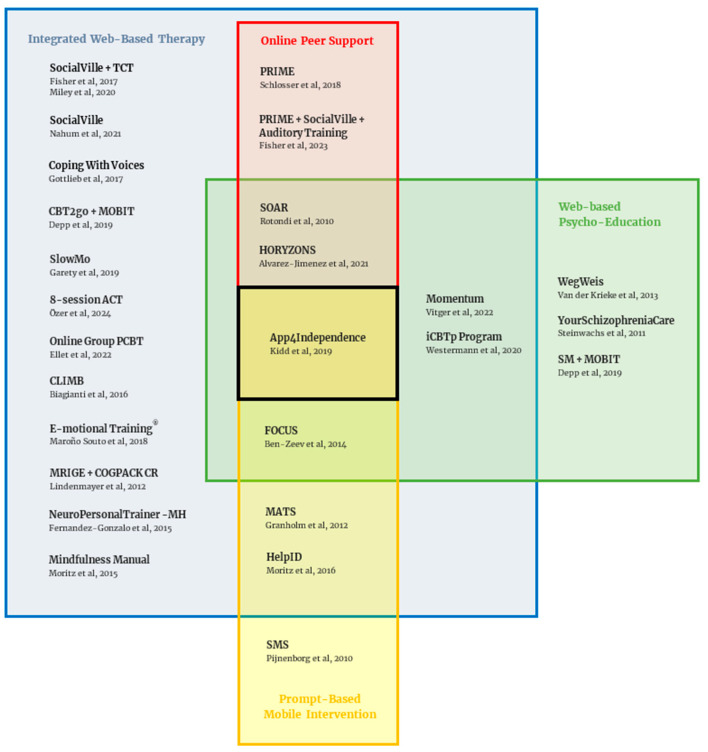
Diagram depicting the overlap of the different intervention types across studies. Abbreviations: TCT (Targeted Cognitive Training), MOBIT (Mobile interaction with a web-based program), CLIMB (Creating Live Interactions to Mitigate Barriers), MRIGE (Mind Reading: Interactive Guide to Emotions), COGPACK CR (Computerized Cognitive Remediation), NPT-MH (NeuroPersonalTrainer-Mental Health), PRIME (Personalized real-time Intervention for Motivational Enhancement), SOAR (Schizophrenia Online Access to Resources), MATS (Mobile Assessment and Treatment for Schizophrenia), SM (Self-Monitoring). References: Nahum et al., 2021 [27]; Fisher et al., 2017 [6], Miley et al., 2020 [21], Gottlieb et al., 2017 [8], Depp et al., 2019 [34], Özer et al., 2024 [46], Ellet et al., 2022 [30], Biagianti et al., 2016 [31], Maroño Souto et al., 2018 [41], Lindenmayer et al., 2012 [25], Fernandez-Gonzalo et al., 2015 [36], Moritz et al., 2015 [26], Schlosser et al., 2018 [40], Fisher et al., 2023 [35], Rotondi et al., 2010 [37], Alvarez Jimenez et al., 2021 [42], Kidd et al., 2019 [32], Ben-Zeev et al., 2014 [20], Granholm et al., 2012 [33], Moritz et al., 2016 [39], Pjinenborg et al., 2010 [29], Vitger et al., 2022 [44], Westermann et al., 2020 [38], Van der Krieke et al., 2013 [23], Steinwachs et al., 2011 [43], Depp et al., 2019 [34].

**Table 2 ijerph-22-01024-t002:** Summary of studies addressing symptom severity.

Interventions with Significant Results	Type	Interventions Without Significant Results	Type
SocialVille + TCT [6,21]	•	SocialVille [27]	•
E-motional Training^®^ [10]	•	HelpID [39]	•
CBT2go + MOBIT [34]	•	CLIMB [31]	•
Coping with Voices [8]	•	MRIGE + COGPACK CR [25]	•
SlowMo [45]	•	NPT-MH [36]	•
iCBTp Program [38]	•	SMS [29]	•
Mindfulness Manual [26]	•	Momentum [44]	••
8-session ACT [46]	•	PRIME [40]	••
Online Group PCBT [30]	•	MATS [33]	••
PRIME + SocialVille + Auditory Training [35]	••		
HORYZONS [42]	•••		
SOAR [37]	•••		
FOCUS [20]	•••		
App4Independence [32]	• • • •		

•—Web-Based Therapy; •—Web-Based Psycho-Education; •—Online Peer Support; •—Web-Based Therapy.

**Table 3 ijerph-22-01024-t003:** Summary of studies addressing depression and anxiety.

Interventions with Significant Results	Type	Interventions Without Significant Results	Type
Mindfulness Manual [26]	•	CBT2go + MOBIT [34]	•
SlowMo [45]	•	Coping with Voices [8]	•
HelpID [39]	•	iCBTp Program [38]	•
FOCUS [20]	•	NPT-MH [36]	•
PRIME + SocialVille + Auditory Training [35]	• •	MATS [33]	• •
Online Group PCBT [30]	• • •	HORYZONS [42]	• • •

•—Web-Based Therapy; •—Web-Based Psycho-Education; •—Online Peer Support; •—Web-Based Therapy

**Table 4 ijerph-22-01024-t004:** Summary of studies addressing functioning.

Interventions with Significant Results	Type	Interventions Without Significant Results	Type
SocialVille [27]	•	NPT-MH [36]	•
SocialVille + TCT [6,21]	•	E-motional Training^®^ [41]	•
SocialVille + PRIME + Auditory Training [35]	• •	SMS [29]	•
CBT2go + MOBIT [34]	•	Momentum [44]	• •
Coping with Voices [8]	•	MATS [33]	• •
8-session ACT [46]	•	PRIME [40]	• •
		HORYZONS [42]	• • •

•—Web-Based Therapy; •—Web-Based Psycho-Education; •—Online Peer Support; •—Web-Based Therapy.

**Table 5 ijerph-22-01024-t005:** Summary of studies addressing motivation.

Interventions with Significant Results	Type	Interventions Without Significant Results	Type
SocialVille + PRIME + Auditory Training [35]	• •	SocialVille [27]	•
PRIME [40]	• •	SMS [29]	•
SocialVille + TCT [6,21]	•	HORYZONS [42]	• • •
iCBTp Program [38]	•		

•—Web-Based Therapy; •—Web-Based Psycho-Education; •—Online Peer Support; •—Web-Based Therapy.

**Table 6 ijerph-22-01024-t006:** Summary of studies addressing mindfulness.

Interventions with Significant Results	Type	Interventions Without Significant Results	Type
SlowMo [45]	•	Momentum [44]	• •
iCBTp Program [38]	•		

•—Web-Based Therapy; •—Web-Based Psycho-Education; •—Online Peer Support; •—Web-Based Therapy.

**Table 7 ijerph-22-01024-t007:** Summary of studies addressing social cognition.

Interventions with Significant Results	Type	Interventions Without Significant Results	Type
SocialVille [27]	•		
SocialVille + TCT [6,21]	•		
iCBTp Program [38]	•		
CLIMB [31]	•		
E-motional Training^®^ [41]	•		
MRIGE + COGPACK CR [25]	•		
NPT-MH [36]	•		

•—Web-Based Therapy; •—Web-Based Psycho-Education; •—Online Peer Support; •—Web-Based Therapy.

**Table 8 ijerph-22-01024-t008:** Summary of studies addressing neurocognition.

Interventions with Significant Results	Type	Interventions Without Significant Results	Type
SocialVille + TCT [6,21]	•	SocialVille [25]	•
MRIGE + COGPACK CR [25]	•	iCBTp Program [38]	•
NPT-MH [36]	•	E-motional Training^®^ [41]	•
SocialVille + PRIME + Auditory Training [35]	• •	MATS [33]	• •

•—Web-Based Therapy; •—Web-Based Psycho-Education; •—Online Peer Support; •—Web-Based Therapy.

**Table 9 ijerph-22-01024-t009:** Summary of studies addressing quality of life.

Interventions with Significant Results	Type	Interventions Without Significant Results	Type
SlowMo [45]	•	SocialVille + TCT [6,21]	•
iCBTp Program [38]	•	HORYZONS [42]	• • •
CLIMB [31]	•	NPT-MH [36]	•
		PRIME [40]	• •

•—Web-Based Therapy; •—Web-Based Psycho-Education; •—Online Peer Support; •—Web-Based Therapy.

**Table 10 ijerph-22-01024-t010:** Summary of studies addressing involvement in treatment.

Interventions with Significant Results	Type	Interventions Without Significant Results	Type
YourSchizophreniaCare [25]	•	WegWeis [23]	•
Momentum [44]	• •	HORYZONS [42]	• • •
SOAR [37]	• • •	FOCUS [20]	• • •
App4Independence [32]	• • • •		
SMS [29]	•		
PRIME [40]	• •		
MATS [33]	• •		

•—Web-Based Therapy; •—Web-Based Psycho-Education; •—Online Peer Support; •—Web-Based Therapy.

## Data Availability

No new data were created or analyzed in this study.

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
