# Peer review of "Mend the Gap: Online User-Led Adjuvant Treatment for Psychosis: A Systematic Review on Recent Findings"

_ijerph, 2025, doi:10.3390/ijerph22071024_

Round 1
Reviewer 1 Report
Comments and Suggestions for Authors
General Comments:
Thank you for the opportunity to review this timely and important manuscript. The topic of user-led, internet-based interventions for schizophrenia spectrum disorders (SSD) is both clinically relevant and socially impactful. The systematic review appears to be thorough and provides a much-needed update to the existing literature in this fast-evolving area. The paper is well-structured, the rationale is clear, and the use of PRISMA guidelines and Cochrane risk assessment strengthens its methodological integrity.
Your focus on treatment gaps—specifically the therapeutic gap—is commendable, as it reflects a deeper understanding of the ongoing needs in SSD management. The writing style is accessible, and the motivation for conducting this review is clearly communicated.
I believe this paper has potential for publication after addressing the following suggestions:
General Remarks:
- Clarity and Flow: The manuscript is comprehensive but could benefit from more concise language in several sections to improve readability.
- Terminology Use: Ensure consistency in terminology (e.g., “psychosis,” “SSD,” “SMI”) to avoid confusion for readers less familiar with the distinctions.
- Recent Literature Integration: While you have included many recent trials, consider adding studies published in 2023–2025 to strengthen the timeliness of the review.
- Narrative Strength: Some narrative transitions between paragraphs could be smoother. Occasionally, new sections begin abruptly without adequate linkage.
- Figures and Tables: Ensure that all figures (e.g., PRISMA and risk of bias) are fully legible and properly described in the text. Add captions that summarize key insights.
- Conclusion Depth: The conclusion could benefit from a stronger emphasis on implications for clinical practice and policy.
- Risk of Bias Interpretation: Consider discussing more clearly how the risk of bias may affect the interpretation of your results.
- Study Overlap: Indicate whether there is participant or data overlap between the included studies.
- Limitations Section: Although implied, a dedicated section on the limitations of your review would enhance transparency and rigor.
- Caregiver and Future-Facing Perspectives: The manuscript would be strengthened by mentioning the role of caregivers in digital interventions for SSDs. Research shows that the quality of life of caregivers, particularly in serious mental illness such as bipolar disorder, is closely tied to the implementation of patient-centered care. A relevant recent article that may help frame this aspect is:
Milic et al. (2025), "The Impact of Patient-Centered Care in Bipolar Disorder: An Opinion on Caregivers’ Quality of Life," Clin. Med., 14, 2209. https://doi.org/10.3390/jcm14072209
Similarly, the discussion could be enriched by incorporating the role of AI and telemedicine in mental health, which is increasingly relevant to online user-led interventions. See:
Milic et al. (2025), "The Role of Artificial Intelligence in Managing Bipolar Disorder: A New Frontier in Patient Care," J. Clin. Med., 14, 2515. https://doi.org/10.3390/jcm14072515
Specific Comments:
- Line 12: Consider clarifying what is meant by “several outcomes.” A brief enumeration here could help orient the reader.
- Lines 18–19: You mention that WBT was the most effective—please clarify what types of WBT interventions were included (e.g., CBT-based? ACT?).
- Line 22: The statement that all 7 studies showed positive outcomes in social cognition is promising—please elaborate on intervention features that may have contributed to this success.
- Line 46: When defining the “treatment gap,” provide a brief clarification of how it differs from the “therapeutic gap” for clearer differentiation.
- Line 66: The phrase “holy grail of treatment research” is impactful but consider providing a citation or rephrasing for a more academic tone.
- Line 73–74: The term “highly specialized” is vague. Could you specify which components (e.g., therapist expertise, technology) contribute to this complexity?
- Line 90–94: The argument about flexibility and reduced training burden is compelling—consider supporting with references to implementation studies or pilot programs.
- Lines 132–134: Your search terms are well chosen, but consider specifying if MeSH terms were used in PubMed.
- Line 154–157: The rationale for excluding studies focused solely on data collection is solid. You might clarify whether any hybrid designs (data collection + intervention) were included.
- Line 266–269: When categorizing studies by outcome, consider adding a summary table in the results section that shows which outcomes were addressed by which studies.
Conclusion:
This is a well-conceived and methodologically sound manuscript that addresses a crucial topic in contemporary mental health care. With revisions aimed at improving clarity, methodological detail, narrative flow, and by incorporating broader caregiving and technological perspectives, it has strong potential to make a meaningful contribution to the literature on digital interventions in SSD.
I hope these suggestions prove helpful.
Comments on the Quality of English LanguagePlease include a native English speaker to improve the language in the revised manuscript
Author Response
We would like to thank the reviewer for the thorough appraisal of our paper, as it has provided ample, actionable feedback on how to improve the relevance and quality of our work. We have employed our best efforts to address each of the recommendations made and, when possible, make the necessary changes, while also preserving our original vision for the article.
To ease reading, we will be addressing the comments point by point, and referencing changes with their corresponding line number, according to the newest version of our article. The proper response to each point is attached below in Word format.

Reviewer 2 Report
Comments and Suggestions for Authors
This is an interesting systematic review involving some online treatments for individuals with Schizophrenia Spectrum disorders. It address an innovative tòpic.
The review is methodologically correct, well-written, and characterized. Although it is a narrative systematic review and no quantitative analysis has been performed, the discussion, limitations, and conclusions are appropriate.
One point that could be dealed is the presentation of the results, as there is a final section for each type of results that outlines the measures used for each grouped aspect. Perhaps it could be presented in a more systematic way in a table or with bullet points to clarify and facilitate reading.
Author Response
Firstly, we would like to thank the reviewer for the helpful feedback on our paper. We have taken into account the overall readability of the article and, although the outcome measure section for each outcome has remained unchanged, we have made changes to improve this aspect:
- We have conducted a read-through of our paper and made the necessary changes in wording and phrasing to streamline the narrative analysis.
- We have included a summary table pertaining to the studies addressing each outcome, as to guide the reader through the "Findings" section of our Outcomes.
As the enumeration of outcome measures was mainly done to emphasize the heterogeneity between the selected papers in how they measure the same outcome, we have chosen to keep the current format for their presentation in our paper.
Reviewer 3 Report
Comments and Suggestions for Authors
I commend the authors for the choice of the topic, thoroughness in their PRISMA systematic approach (checklist fully addressed). It's refreshing to read such a thorough review. If the number of words is suitable for the journal I fully support the publication of this paper in its format.
That said, I would like to offer a few suggestions for consideration:
1. The goal of the paper is very broad. I would have selected specific variables, such as social cognition, positive symptoms, or attention/memory. I am aware that the literature in this field is very broad. I would have added additional selection criteria. I am aware that this change would lead to restarting this review from scratch. I will let you decide whether your journal can accommodate such a large broad paper. The paper is well written and technically sound, so it cannot be rejected per se.
2. The authors of this paper have indeed addressed all the PRISMA criteria. However, I would have added extra exclusion/inclusion criteria to make this paper briefer but with more impact.
Author Response
Firstly we would like to thank the reviewer for the helpful feedback provided. We will be responding to the recommendations made through the MDPI platform, as well as those additionally shared by the editor via email:
Adressing the recommentations visible in the MDPI platform:
We agree that our paper casts a broad net over the burden of SSD, analysing many outcomes, some more promising than others. This can indeed make it more difficult for the general readablility of our article, as the most important conclusions can become "buried" under less relevant information. To adress this, specifically in the Results section, we have added 10 new tables, each of them summarizing the studies addressing each outcome, categorizing them according to the significance of their findings, as well as their intervention type, through color-coding that is consistent throughout the paper. We believe that, with these changes, the article has become more segmented and easier to understand, defering the need for a major restructuring of the overall work.
Adressing the additional recommentations shared via email:
"... the authors have been extremely verbose. As a result, the paper loses its scientific impact and may be confusing for researchers that are not in this research field. Hence, I queried the number of words as this would be problematic."
Regarding this point, we believe that the new adittion of summary tables in each outcome section serves to alleviate the difficulties in reading through the narrative analysis. They can provide a guide to the reader as they go through the thorough, yet necessary, in our opinion, overview of the included studies' findings.
"Regarding the conclusion’s consistency with the evidence and arguments presented and
addressing the main question posed, it gets lost in the results."
We have added to our Conclusions section in order to highlight the main finding we would like to communicate with this paper: the promising results found in Social Cognition and the possibility of targeting this outcome through online user-led interventions.
"It's a review so figures are not of relevance and current tables are probably too full of
information but ok"
We have aimed to provide readers with an acessible, yet comprehensive depiction of our results, in a way that each of the included studies' design, goal and results, can be easily recalled whenever needed, as the reader progresses through the narrative portion. As such, we believe the current state of our tables is satisfactory.
We look forward to hear feedback on the newest changes made, in order to further improve our paper before it is hopefully published.
Best regards.